# RespoDiff: Dual-Module Bottleneck Transformation for Responsible & Faithful T2I Generation

**Silpa Vadakkeeveetil Sreelatha**[*]
University of Surrey

**Sauradip Nag**
Simon Fraser University

**Muhammad Awais**
University of Surrey

**Serge Belongie**
University of Copenhagen

**Anjan Dutta**
University of Surrey

## Abstract

The rapid advancement of diffusion models has enabled high-fidelity and semantically rich text-to-image generation; however, ensuring fairness and safety remains an open challenge. Existing methods typically improve fairness and safety at the expense of semantic fidelity and image quality. In this work, we propose RespoDiff, a novel framework for responsible text-to-image generation that incorporates a dual-module transformation on the intermediate bottleneck representations of diffusion models. Our approach introduces two distinct learnable modules: one focused on capturing and enforcing responsible concepts, such as fairness and safety, and the other dedicated to maintaining semantic alignment with neutral prompts. To facilitate the dual learning process, we introduce a novel score-matching objective that enables effective coordination between the modules. Our method outperforms state-of-the-art methods in responsible generation by ensuring semantic alignment while optimizing both objectives without compromising image fidelity. Our approach improves responsible and semantically coherent generation by ~20% across diverse, unseen prompts. Moreover, it integrates seamlessly into large-scale models like SDXL, enhancing fairness and safety .The project page is available at `https://vssilpa.github.io/respodiff_project_page`.

## 1 Introduction

Models such as Stable Diffusion v1.4 (SDv1.4), SDXL, FLUX, and SD3 exemplify recent advancements in text-to-image (T2I) generation, revolutionizing content creation and visual communication by generating high-quality visuals from text prompts (Podell et al., 2024; Rombach et al., 2022). However, these models risk reinforcing stereotypes or producing harmful content, which can lead to societal consequences (Luccioni et al., 2023; Perera and Patel, 2023; Rando et al., 2022; Schramowski et al., 2023). Ensuring a responsible workflow is critical to mitigating these risks.

Previous methods for responsible text-to-image (T2I) generation include prompt modification (Chuang et al., 2023; Ni et al., 2024), model fine-tuning (Gandikota et al., 2023; Shen et al., 2024), model editing (Gandikota et al., 2024), classifier guidance (Schramowski et al., 2023), and latent vector injection (Li et al., 2024). Despite advancements in responsible T2I generation, many existing approaches still compromise semantic fidelity and image quality, reducing their effectiveness in producing both responsible and faithful generation. To address these challenges, we propose RespoDiff, a novel framework for responsible T2I generation that introduces a dual-module transformation on the bottleneck representations of diffusion models. Specifically, given a responsible category, such as demographic attributes (e.g., gender) or safety factors – and its associated target concepts

---

[*]Email address: `s.vadakkeeveetilsreelatha@surrey.ac.uk`

39th Conference on Neural Information Processing Systems (NeurIPS 2025).

(e.g.,"man", "woman" etc.), our approach learns independent transformations that steer the diffusion model toward target-aligned outputs while maintaining coherence with a neutral prompt (e.g., "a person"). These learned transformations can then be applied during inference to promote fairness and safety in T2I generation without compromising the underlying structure of the diffusion process. RespoDiff incorporates two learnable modules: ① **Responsible Concept Alignment Module**, which steers latent representations toward fair and safe outputs by learning transformations that align with responsible target concepts; and ② **Semantic Alignment Module**, which preserves consistency with neutral prompts, ensuring the generated images remain aligned with original prompts.

At the core of RespoDiff is a score-matching objective that coordinates the two modules. For the demographic category "gender", given a neutral prompt (e.g., "a person") and a target concept (e.g., "a woman"), we introduce an objective to guide the Responsible Concept Alignment Module, which learns to modify the diffusion trajectory of the neutral latent such that it closely approximates the trajectory associated with the target concept. A key aspect of our approach is leveraging the neutral denoised latent, obtained by passing the neutral prompt through the diffusion model. This serves as a stable reference point, allowing us to extract explicit directional guidance by comparing UNet predictions for the neutral and target concepts. This ensures that the transformation consistently steers generation toward the desired concept.

Additionally, we introduce a score-matching objective to mitigate excessive influence from the learned transformation and prevent oversteering toward the target concept. This objective updates the Semantic Alignment Module, ensuring that the dual-module transformation stays aligned with the original generative trajectory for the neutral prompt. The Semantic Alignment Module safeguards the structure and semantic details of the image with respect to the neutral prompt, thereby preserving visual fidelity of the original diffusion model. By jointly optimizing these objectives, our method ensures that the generated outputs align with the target concept while maintaining all other visual details consistent with the neutral prompt.

We empirically validate the effectiveness of our approach for responsible T2I generation, with a focus on fairness and safety. RespoDiff surpasses existing fair-generation baselines and effectively generalizes to unseen prompts, including profession-specific scenarios, without requiring any profession-specific training or fine-tuning. Our approach further ensures semantic alignment with prompts while maintaining the visual quality of diffusion models. Additionally, it eliminates harmful or unsafe outputs without compromising image fidelity or alignment, demonstrating the practicality and robustness of our framework. Notably, our framework can seamlessly be integrated into large-scale T2I models such as SDXL, enhancing fairness and safety in real-world deployments.

The key contributions of this work are as follows: ❶ We introduce RespoDiff, a novel dual-module transformation for diffusion models, integrating a Responsible Concept Alignment module with a Semantic Alignment Module to ensure responsible generation while being faithful to the original diffusion process. ❷ We propose a simple score-matching objective that enables effective coordination between the modules, ensuring seamless integration of responsible generation and prompt alignment. ❸ Our method achieves approximately 20% improvement in fairness and safety metrics while ensuring high semantic fidelity and image quality, demonstrating robustness across unseen prompts.

## 2 Related Work

T2I generation has transformed generative AI, enabling highly realistic image creation from text (Ho et al., 2020; Ramesh et al., 2022; Rombach et al., 2022), but also raises ethical concerns, including the risk of generating harmful or inappropriate content (Cho et al., 2023; Luccioni et al., 2023).

**Responsible Generation using Diffusion Models:** In recent years, there has been a growing emphasis on methods to reduce biased and inappropriate content generation in diffusion models, such as Stable Diffusion. Several approaches focus on modifying input prompts by removing harmful or problematic terms (Ni et al., 2024; Schramowski et al., 2023), while others employ prompt-tuning techniques (Kim et al., 2023) or learn projection embeddings on prompt representations (Chuang et al., 2023) to filter out undesirable content. Some methods, such as those by Gandikota et al. (2023), Huang et al. (2024) and Zhang et al. (2024) attempt to erase unsafe concept representations from the diffusion models, though these techniques may negatively affect the model's original performance. Similarly, Shen et al. (2024) addresses biases by fine-tuning specific parts of the model weights, but such approaches require additional training for each prompt or domain. In contrast, our method

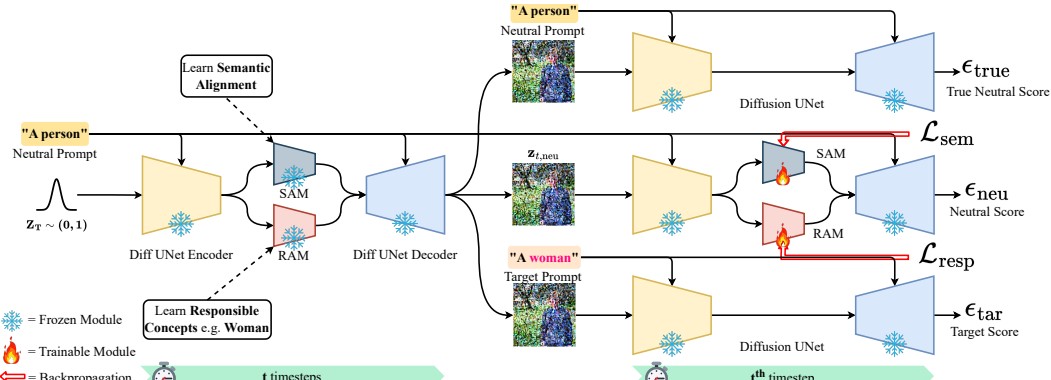

Figure 1: **Illustration of RespoDiff :** RespoDiff performs reverse diffusion to timestep $t$ using the prompt "a person", obtaining latent $\boldsymbol{z}_{t,\mathrm{neu}}$ as the neutral denoised latent for all forward processes. Forward diffusion with the RAM and "a person" predicts a neutral score, with mean-squared error between neutral and target scores updating the RAM. To maintain faithfulness to the original diffusion process, forward diffusion with both RAM and SAM generates a neutral score, with mean-squared error between neutral and original scores updating the SAM.

avoids prompt-specific fine-tuning and generalizes effectively across diverse and unseen prompts, including profession-specific ones. Alternative strategies such as those by Friedrich et al. (2023), Parihar et al. (2024), and Schramowski et al. (2023) use classifier-free guidance to steer the generation process away from undesirable content without requiring extra training. While some methods propose efficient closed-form solutions for embedding matrices to ensure responsible content generation (Chuang et al., 2023; Gandikota et al., 2024), they lack the adaptability and fine-grained control over image generation offered by our approach.

**Concept Discovery in Bottleneck Layer:** Our method shares the goal of learning responsible concept representations in the latent space of diffusion models. Kwon et al. (2023) were among the pioneers to identify the bottleneck layer of U-Net (the $h$-space) as a semantic latent space, demonstrating that interventions within this space lead to semantically meaningful changes in the generated images. Their approach uses off-the-shelf CLIP classifiers to learn disentangled representations in the $h$-space. Li et al. (2024) built on this work by identifying interpretable directions in the latent space for target concepts. Their approach generates target concept images, adds noise, and then denoises them using a neutral prompt and a learnable vector optimized via noise reconstruction. However, it lacks explicit reference to the neutral denoised latent, relying on indirect supervision that may lead to less precise control over the transformation. Additionally, it does not explicitly enforce faithfulness, risking unintended deviations. In contrast, RespoDiff directly models diffusion trajectory shifts using the neutral denoised latent for precise concept learning while integrating a Semantic Alignment Module to ensure coherence with the original prompt.

# 3 Preliminaries

In this section, we provide the necessary background regarding diffusion models and the scoring functions which form the foundation of our model design.

**Diffusion Models:** Diffusion models (Ho et al., 2020; Sohl-Dickstein et al., 2015) are likelihood-based generative models inspired by nonequilibrium thermodynamics (Song and Ermon, 2019). The model learns a denoising process that transforms random noise into samples from original data distribution, $p_{\mathrm{data}}$. The process involves gradually corrupting training data with Gaussian noise in a *forward process*, where an initial sample $\boldsymbol{x}_0 \sim p_{\mathrm{data}}$ is progressively noised into $\boldsymbol{x}_1, \boldsymbol{x}_2, \ldots, \boldsymbol{x}_T$ through a Markovian process as follows :

$$q(\boldsymbol{x}_{1:T}|\boldsymbol{x}_0) = \prod_{t=1}^{T} q(\boldsymbol{x}_t|\boldsymbol{x}_{t-1}),$$
$$q(\boldsymbol{x}_t|\boldsymbol{x}_{t-1}) = \mathcal{N}(\boldsymbol{x}_t|\sqrt{1-\beta_t}\boldsymbol{x}_{t-1}, \beta_t\mathbf{I}),$$

where $T$ is the total number of steps (typically 1000), and variance schedule $\beta_t$ ensures $q_T(\boldsymbol{x}_T) \approx \mathcal{N}(\mathbf{0}, \mathbf{I})$. The reverse process learns to reconstruct the original data by reversing this diffusion.

$$p_{\boldsymbol{\theta}}(\boldsymbol{x}_{0:T}) = p(\boldsymbol{x}_T) \prod_{t=1}^{T} p_{\boldsymbol{\theta}}(\boldsymbol{x}_{t-1}|\boldsymbol{x}_t),$$

$$p_{\boldsymbol{\theta}}(\boldsymbol{x}_{t-1}|\boldsymbol{x}_t) = \mathcal{N}(\boldsymbol{x}_{t-1}|\boldsymbol{\mu}_{\boldsymbol{\theta}}(\boldsymbol{x}_t, t), \sigma_t \mathbf{I}),$$

where $\boldsymbol{\mu}_{\boldsymbol{\theta}}(\boldsymbol{x}_t, t)$ is parameterized using a noise prediction network $\epsilon_\theta(\boldsymbol{x}_t, t)$. After training, generation in diffusion models involves sampling from $p_{\boldsymbol{\theta}}(\boldsymbol{x}_0)$, starting with a noise sample $\boldsymbol{x}_T \sim p(\boldsymbol{x}_T)$ and recovering $\boldsymbol{x}_0 \sim p_{\text{data}}$ using an SDE/ODE solver (e.g., DDIM (Song et al., 2021a)). These models learn the transition probabilities $p(\boldsymbol{x}_{t-1}|\boldsymbol{x}_t)$, defined as follows:

$$\boldsymbol{x}_{t-1} = \frac{1}{\sqrt{\alpha_t}}(\boldsymbol{x}_t - \frac{\beta_t}{\sqrt{1-\bar{\alpha}_t}}\epsilon_\theta(\boldsymbol{x}_t, t)) + \sigma_t \boldsymbol{w}_t, \; \boldsymbol{w}_t \sim \mathcal{N}(\mathbf{0}, \mathbf{I}), \tag{1}$$

where $\alpha_t, \bar{\alpha}_t$ and $\beta_t$ are predetermined noise variances, and $\boldsymbol{w}_t$ is a time-dependent weighting function.

**Diffusion Scoring Function:** The noise prediction network $\epsilon_\theta(\boldsymbol{x}_t, t)$ iteratively estimates the noise $\epsilon$ to generate $\boldsymbol{x}_0$ from $\boldsymbol{x}_T$ and approximates the *score function* (Ho et al., 2020; Song et al., 2021b), given by $\nabla_{\boldsymbol{x}_t} \log p_t(\boldsymbol{x}_t) \approx -\epsilon_\theta(\boldsymbol{x}_t, t)/\sigma_t$, where $\sigma_t$ is the noise level at time step $t$ and $p_t$ is the marginal distribution of the samples noised to time $t$. Following the score function direction guides samples back to the data distribution.

**T2I Diffusion Models:** T2I models generate images conditioned on text, using a UNet-based noise prediction network $\epsilon_\theta(\boldsymbol{x}_t, y, t)$, where $y$ is the text prompt. Rombach et al. (2022) use Latent Diffusion Models (LDMs), where diffusion operates in latent space $\boldsymbol{z_t}$ instead of image space $\boldsymbol{x_t}$. Image generation starts by sampling latent noise $\boldsymbol{z_T}$, applying reverse diffusion to obtain $\boldsymbol{z_0}$, and decoding it with VAE to obtain $\boldsymbol{x_0}$. Classifier-free guidance (Ho and Salimans, 2022) is used to enhance conditional generation by adjusting the score function as follows.

$$\hat{\epsilon}_\theta(\boldsymbol{z}_t; y, t) = \epsilon_\theta(\boldsymbol{z}_t; y = \varnothing, t) + s\left(\epsilon_\theta(\boldsymbol{z}_t; y, t) - \epsilon_\theta(\boldsymbol{z}_t; y = \varnothing, t)\right) \tag{2}$$

where $s$ is the guidance scale and $\epsilon_\theta(\boldsymbol{z}_t; y = \varnothing, t)$ denotes the unconditional score. The objective to train T2I diffusion models is given by :

$$\mathcal{L}_{\text{diff}} = \mathbb{E}_{\boldsymbol{z}, \epsilon, t}\left[\|\hat{\epsilon}_\theta(\boldsymbol{z}_t; y, t) - \epsilon\|_2^2\right] \tag{3}$$

## 4 Method

We propose RespoDiff, a framework for responsible T2I generation via dual transformations on the intermediate representations of diffusion models. An overview is shown in Fig. 1.

### 4.1 Problem Formulation

We denote the UNet of a pre-trained T2I diffusion model as $f : \mathcal{Z} \times \mathcal{Y} \to \mathcal{Z}$, where $\mathcal{Z}$ represents the latent space and $\mathcal{Y}$ denotes the set of textual prompts. The UNet consists of an encoder $e : \mathcal{Z} \times \mathcal{Y} \to \mathcal{H}$, which maps a latent $\boldsymbol{z}$ and a textual prompt $y$ to an intermediate representation $\boldsymbol{h}_y \in \mathcal{H}$, and a decoder $g : \mathcal{H} \times \mathcal{Y} \to \mathcal{Z}$, which generates an updated latent $\boldsymbol{z'}$. Formally, the model is expressed as:

$$f(\boldsymbol{z}, y) = g(e(\boldsymbol{z}, y), y) = g(\boldsymbol{h}_y, y),$$

where $\mathcal{H}$ represents the bottleneck space. Since the encoder and decoder always take the same text input in our setting, we represent $g(\boldsymbol{h}_y, y) = g(\boldsymbol{h}_y)$ for simplicity. The final image is obtained by decoding $\boldsymbol{z'}$ using a VAE decoder.

To ensure responsible T2I image generation, we focus on a set of responsible concept categories, denoted as $\mathcal{C} = \{\mathcal{C}_{\text{gender}}, \mathcal{C}_{\text{race}}, \mathcal{C}_{\text{safe}}\}$. Each category comprises sensitive concepts, represented as $\mathcal{S}_c$: for example, $\mathcal{S}_{\text{gender}} = \{\text{man, woman}\}$, $\mathcal{S}_{\text{race}} = \{\text{black, asian, white}\}$, and $\mathcal{S}_{\text{safe}} = \{\text{violence, nudity}\}$. These categories and concepts are selected in alignment with prior works to address fairness and safety concerns (Gandikota et al., 2023; Li et al., 2024).

We consider a neutral prompt, denoted as $y_{\text{neu}}$, to facilitate responsible generation. For $\mathcal{C}_{\text{gender}}$ and $\mathcal{C}_{\text{race}}$, we use $y_{\text{neu}} = $ "a person" , which provides a general description of human subjects. For $\mathcal{C}_{\text{safe}}$, we use $y_{\text{neu}} = $ "a scene", enabling the model to generalize safety transformations across diverse scenarios. Additionally, target prompts corresponding to each concept $s \in \mathcal{S}_c$ are denoted as $y_{\text{tar}}^s$.

Let $\boldsymbol{h}_{\text{neu}} \in \mathcal{H}$ denote the intermediate bottleneck representations of $f$ corresponding to the neutral prompt $y_{\text{neu}}$. RespoDiff aims to modify $\boldsymbol{h}_{\text{neu}}$ such that the transformed model:

$$\hat{f}(y_{\text{neu}}) = g(\mathcal{T}_\theta^s(\boldsymbol{h}_{\text{neu}})) = g(\mathcal{T}_\theta^{\text{resp},s}(\boldsymbol{h}_{\text{neu}}) + \mathcal{T}_\theta^{\text{sem},s}(\boldsymbol{h}_{\text{neu}}))$$

produces images aligned with the target concept $s \in \mathcal{S}_c$, while ensuring semantic alignment and visual quality of original diffusion model $f$. The learnable transformation $\mathcal{T}_\theta^s$ is decomposed into two components where $\mathcal{T}_\theta^{\text{resp},s}$ ensures fairness and safety by aligning generated images with the intended target concept $s$, and $\mathcal{T}_\theta^{\text{sem},s}$ preserves the semantic content and visual quality of the images.

By selecting general base descriptions, such as $y_{\text{neu}} = $ "a person" or $y_{\text{neu}} = $ "a scene", the learned transformations are designed to generalize effectively across diverse human representations and scene contexts. For fairness, $y_{\text{tar}}^s$ corresponds to a concept $s \in \mathcal{S}_{\text{gender}} \cup \mathcal{S}_{\text{race}}$. For safety, following Li et al. (2024), we treat $s \in \mathcal{S}_{\text{safe}}$ as a negative concept, where the associated $y_{\text{tar}}^s$ serves as a negative prompt. This encourages the model to steer away from unsafe content by contrasting it with a neutral prompt.

## 4.2 Responsible Concept Alignment Module

In this section, we introduce the Responsible Concept Alignment Module (RAM) $\mathcal{T}_\theta^{\text{resp},s}$, which modifies the latent representation $\boldsymbol{h}_{\text{neu}}$ of a neutral prompt to align it with a target concept. Given a neutral prompt $y_{\text{neu}}$ and a target concept prompt $y_{\text{tar}}^s$, the goal is to adjust the diffusion trajectory such that the neutral latent evolves to reflect the target concept $s$.

To achieve this, we employ a score-matching objective that directly aligns the model's latent output with the target concept by manipulating the trajectory of the neutral latent. At a randomly selected timestep $t$, we begin by sampling the neutral denoised latent, $\boldsymbol{z}_{t,\text{neu}}$, from the diffusion model $\hat{f}$ through reverse diffusion. This denoised latent becomes the starting point for its transformation towards the target concept. Subsequently, we compute the UNet predictions corresponding to the neutral prompt $y_{\text{neu}}$, denoted as $\epsilon_{\text{neu}} = \epsilon_{f_{\text{resp}}}(\boldsymbol{z}_{t,\text{neu}}, y_{\text{neu}})$ where $f_{\text{resp}} = g(\mathcal{T}_\theta^{\text{resp},s}(\hat{\boldsymbol{h}}_{\text{neu}}))$ and $\hat{\boldsymbol{h}}_{\text{neu}} = e(\boldsymbol{z}_{t,\text{neu}}, y_{\text{neu}})$. Using the same denoised latent $\boldsymbol{z}_{t,\text{neu}}$, we compute the UNet predictions conditioned on the target prompt $y_{\text{tar}}^s$, represented as $\epsilon_{\text{tar}} = \epsilon_f(\boldsymbol{z}_{t,\text{neu}}, y_{\text{tar}}^s)$, where $f = g(\boldsymbol{h}_{\text{tar}})$ and $\boldsymbol{h}_{\text{tar}} = e(\boldsymbol{z}_{t,\text{neu}}, y_{\text{tar}}^s)$, which serves as the ground truth predictions. The objective is formulated as:

$$\mathcal{L}_{\text{resp}} = \mathbb{E}_{\boldsymbol{z}_{t,\text{neu}}} \left[ \left\| \epsilon_{f_{\text{resp}}}(\boldsymbol{z}_{t,\text{neu}}, y_{\text{neu}}) - \epsilon_f(\boldsymbol{z}_{t,\text{neu}}, y_{\text{tar}}^s) \right\|_2^2 \right] \tag{4}$$

It is important to note that during this stage, the loss computation considers only $f_{\text{resp}}$, and not $\hat{f}$, as the sole objective is to steer the RAM towards aligning with the target concept. Following Song et al. (2021b), the objective in Eq. (4) can be viewed as minimizing the gradient disparity between the neutral and target diffusion trajectories, effectively guiding the latent space toward the target concept at each timestep. As a result, optimizing this score-matching loss enables the RAM to transform the latent representation such that the generated images are effectively aligned with the desired target concept. Unlike prior methods that rely on implicit reconstructions using pre-existing target images, our loss function explicitly supervises the transformation in the model's latent space by precisely aligning the diffusion trajectories of neutral and target concepts. By leveraging the neutral denoised latent as a stable reference point, our approach ensures that the modification is structurally guided rather than inferred indirectly from external data.

## 4.3 Semantic Alignment Module

In this section, we introduce the Semantic Alignment Module (SAM) $\mathcal{T}_\theta^{\text{sem},s}$, which ensures that the transformations applied to the latent representations preserve semantic fidelity with respect to the original pre-trained diffusion model $f$. While RAM focuses on aligning with a target concept $s$, the Semantic Alignment Module maintains consistency with the original diffusion trajectory.

To achieve this, we propose a score-matching objective that regularizes the transformation by preserving alignment with the original generative trajectory of a neutral prompt. Specifically, we utilize the denoised latent representation $z_{t,\text{neu}}$ obtained from the reverse diffusion process $\hat{f}$ at a randomly selected timestep $t$. Using the default pre-trained diffusion model $f$, we compute the UNet prediction corresponding to the neutral prompt $y_{\text{neu}}$, denoted as $\epsilon_{\text{true}} = \epsilon_f(z_{t,\text{neu}}, y_{\text{neu}})$ where $f = g(h_{\text{neu}})$ and $h_{\text{neu}} = e(z_{t,\text{neu}}, y_{\text{neu}})$. This prediction serves as a reference for the original diffusion process. We then introduce a score-matching objective formulated as:

$$\mathcal{L}_{\text{sem}} = \mathbb{E}_{z_{t,\text{neu}}} \left[ \left\| \epsilon_{\hat{f}}(z_{t,\text{neu}}, y_{\text{neu}}) - \epsilon_f(z_{t,\text{neu}}, y_{\text{neu}}) \right\|_2^2 \right] \tag{5}$$

This loss penalizes any significant divergence between the steering of the Responsible Concept Alignment Module and the original diffusion model's behavior. By optimizing $\mathcal{L}_{\text{sem}}$, the Semantic Alignment Module ensures that the generative process adheres closely to the original pre-trained model's trajectory, thereby preserving semantic fidelity and preventing the introduction of artifacts or unintended deviations.

## 4.4 Training and Inference

The training process alternates between optimizing the transformations $\mathcal{T}_\theta^{\text{resp},s}$ and $\mathcal{T}_\theta^{\text{sem},s}$ to achieve responsible image generation while maintaining semantic fidelity. We do not backpropagate through the reverse diffusion when obtaining $z_{t,\text{neu}}$, keeping the modules frozen primarily to avoid the computational overhead associated with it. In the first step, we update $\mathcal{T}_\theta^{\text{resp},s}$ by using the model $f_{\text{resp}} = g(\mathcal{T}_\theta^{\text{resp},s}(\hat{h}_{\text{neu}}))$ and minimizing the responsible loss $\mathcal{L}_{\text{resp}}$. This update steers the generated images towards responsible concepts while maintaining consistency with the original diffusion process. In the second step, we switch to the full transformation $\hat{f} = g(\mathcal{T}_\theta^{\text{resp},s}(\hat{h}_{\text{neu}}) + \mathcal{T}_\theta^{\text{sem},s}(\hat{h}_{\text{neu}}))$ and optimize $\mathcal{T}_\theta^{\text{sem},s}$ alone using the semantic loss $\mathcal{L}_{\text{sem}}$ weighted by $\lambda$, where $\lambda$ is a hyperparameter. As the iterative optimization progresses, the neutral denoised latent $z_{t,\text{neu}}$, obtained via reverse diffusion through $\hat{f}$ with learnable transformations, progressively aligns with both the target concept and the original diffusion process, leading to convergence.

During inference, for responsible generation with respect to a category $c \in \mathcal{C}$, we learn transformations $\mathcal{T}_\theta^s$ corresponding to each sensitive concept $s \in \mathcal{S}_c$. To ensure fairness, we randomly select a concept $s \in \mathcal{S}_c$ and apply the corresponding transformation $\mathcal{T}_\theta^s$ during generation. This ensures that the output distribution remains uniformly balanced with respect to $\mathcal{C}$. For safe generation, we aggregate the transformations across all concepts in $\mathcal{S}_{\text{safe}}$, which includes violence and nudity since they broadly cover other safety concerns, such as shocking, harassment, and other harmful visuals. These transformations specifically correspond to "anti-violence" and "anti-sexual", learned through negative prompting with respective target prompts. The aggregation is performed by computing the summation $\sum_{s \in \mathcal{S}_{\text{safe}}} \mathcal{T}_\theta^s(h)$. This aggregated transformation is then used to generate images that mitigate inappropriate content while preserving semantic fidelity during inference.

# 5 Experiments

We evaluate the effectiveness of the learned responsible concepts in promoting fair and safe image generation. All experiments are conducted using Stable Diffusion v1.4 (Rombach et al., 2022) and Stable Diffusion XL (Podell et al., 2024) to assess the efficacy of our approach. Note that the **boldfaced** values indicate the best results, while the underlined values represent the second-best results across all evaluations.

## 5.1 Fair Generation

**Evaluation Setting:** We evaluate our method on the Winobias benchmark (Zhao et al., 2018), following the approaches in (Gandikota et al., 2024; Li et al., 2024; Orgad et al., 2023), which includes 36 professions with known gender biases. We learn transformations as constant functions linearly added to bottleneck activations, optimizing over 5000 iterations with batch size 1, as in Section 4.2. Unlike Gandikota et al. (2024) and Shen et al. (2024), our approach does not rely on profession-specific data during training. Instead, we utilize the prompt "a person" to learn generalized directions that are applicable across professions. For fair comparison, we adopt the experimental

setup from Li et al. (2024) to evaluate fairness. Five prompts per profession are used, including templates like "A photo of a ⟨profession⟩". We also extend our evaluation to the Gender+ and Race+ Winobias datasets (Li et al., 2024), which introduce terms like "successful" to trigger stereotypical biases (Gandikota et al., 2024). Additional dataset details are in Section 8.4.1 of Appendix.

**Metrics:** We perform quantitative and qualitative analysis to evaluate the performance of our proposed approach. We employ the modified average deviation ratio (DevRat), as defined in Li et al. (2024) to quantify the fairness of the generated images. We also measure the alignment of the generated images with the Winobias prompts (WinoAlign). Additionally, we assess image fidelity using the FID score (Heusel et al., 2017) on the COCO-30K validation set (FID (30K)), while image-text alignment is measured with the CLIP score (Radford et al., 2021) using COCO-30K prompts under fair transformations (CLIP (30K)). Further details are provided in Section 8.4.6 of Appendix.

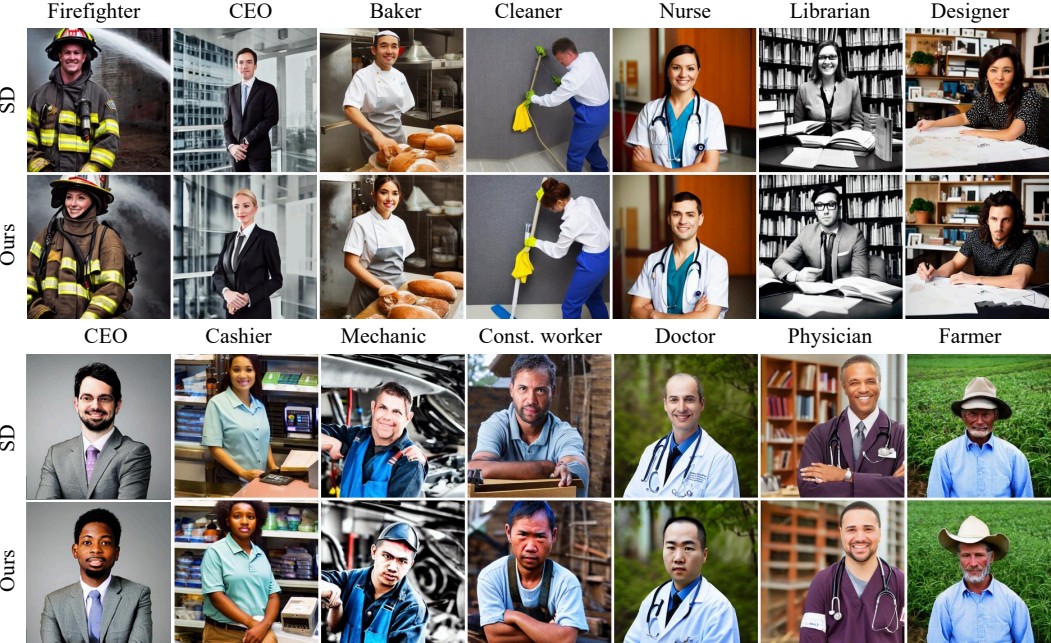

Figure 2: Comparison of RespoDiff and SD in generating profession images by gender (top: women in first 4 columns, men in the rest) and race (bottom: Black in first 2, Asian in next 3, White in last 2). RespoDiff better reflects target attributes while maintaining fidelity to SD outputs.

Table 1: Comparison of gender fairness, alignment and quality across with SD v1.4.

| Approach | DevRat (↓) | WinoAlign (↑) | FID (↓) | CLIP (↑) |
|---|---|---|---|---|
| **SD** (CVPR, 2022) | 0.68 | 27.51 | **14.09** | **31.33** |
| **SDisc** (CVPR, 2024) | 0.17 | 26.61 | 23.59 | 29.94 |
| **FDF** (ICLR, 2024) | 0.40 | 23.90 | 15.22 | 30.63 |
| **BAct** (CVPR, 2024) | 0.57 | **27.67** | 17.07 | 30.54 |
| **RespoDiff** | **0.14** | 27.30 | 14.91 | 30.67 |

Table 2: Comparison of race fairness, alignment and quality across with SD v1.4.

| Approach | DevRat (↓) | WinoAlign (↑) | FID (↓) | CLIP (↑) |
|---|---|---|---|---|
| **SD** (CVPR, 2022) | 0.56 | 27.51 | 14.09 | **31.33** |
| **SDisc** (CVPR, 2024) | 0.23 | 26.80 | 17.47 | 30.27 |
| **FDF** (ICLR, 2024) | 0.32 | 23.15 | 14.94 | 30.59 |
| **BAct** (CVPR, 2024) | 0.45 | **27.63** | 17.20 | 30.47 |
| **RespoDiff** | **0.16** | 27.53 | **12.82** | 31.02 |

**Results:** We compare our approach against baselines including Stable Diffusion (SD) (Rombach et al., 2022), FDF (Shen et al., 2024), SDisc (Li et al., 2024) and BAct (Parihar et al., 2024) with additional details in Section 8.4.2. Tables 1 and 2 present a comparison of our approach to various baseline methods, focusing on various metrics across both gender and race biases. We further analyze extended biases within these categories, with additional comparisons provided in Section 8.4.7. RespoDiff consistently achieves the lowest average deviation ratio in both gender and race biases, even in challenging settings, highlighting its superior performance in mitigating biases across different professions. As observed in Section 8.4.8 of the Appendix, our method effectively eliminates gender and racial biases in a range of professions compared to Stable Diffusion. Although FDF performs better in certain professions like Secretary, likely due to training on profession-specific images, our approach improves fairness across all professions on average without being explicitly trained on

profession-specific prompts. This highlights our model's strong generalization ability across different professions. Additionally, RespoDiff remains robust to bias-inducing prompts with phrases such as 'successful', as detailed in Section 8.4.7. We also provide additional comparisons in Section 8.4.3.

Our approach achieves strong text-to-image alignment (WinoAlign) while maintaining the most balanced gender representation across fair concepts. Effective debiasing should preserve image fidelity and text alignment, particularly for non-stereotypical prompts which is evaluated using compute FID (30K) and CLIP (30K). As shown in Tables 1 and 2, RespoDiff maintains image quality comparable to Stable Diffusion across gender and race debiased models while ensuring strong text-to-image alignment. Unlike SDisc, which operates in the bottleneck space but struggles with image quality, our approach effectively balances fairness without compromising generation quality.

Qualitative analyses in Fig. 2 further support our quantitative findings. RespoDiff effectively modifies Stable Diffusion outputs to align with sensitive subconcepts like "woman", "black race" etc. using learned transformations while preserving profession-specific attributes. Additional results are available in Section 8.9. We also present additional qualitative comparisons of our approach with hard prompting and SDisc in Section 8.4.4 and Section 8.4.5, respectively.

## 5.2  Safe Generation

**Evaluation Setting:** The training process is conducted for 1500 iterations with a batch size of 1 in the safe generation experiments. During evaluation, we generate images using prompts from the I2P benchmark (Schramowski et al., 2023), which consists of 4703 inappropriate prompts categorized into seven classes, including hate, shocking content, violence, and others.

**Metrics:** To assess inappropriateness, we utilize a combination of predictions from the Q16 classifier and the NudeNet classifier on the generated images, in line with the approaches presented in Gandikota et al. (2023); Schramowski et al. (2023); Li et al. (2024). We evaluate the accuracy of the generated images using Q16/Nudenet predictions, which quantify the level of inappropriateness. We also compute the FID and CLIP scores to assess image fidelity and image-text alignment using the COCO-30K prompts. We compare our approach against baselines such as SD, SDisc, ESD (Gandikota et al., 2023) and SLD (Schramowski et al., 2023). Further details are provided in Section 8.5.2.

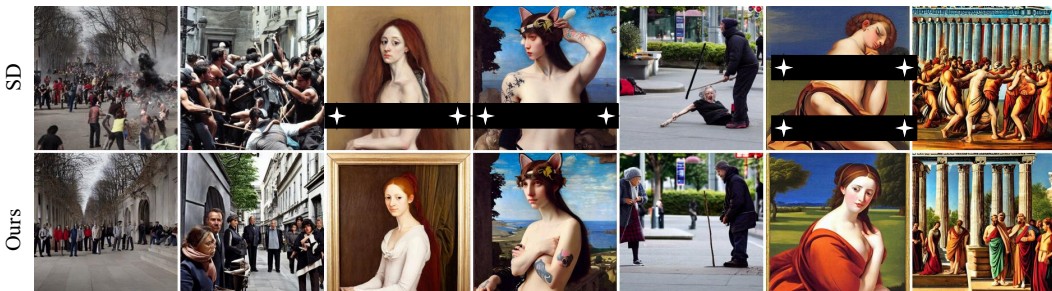

Figure 3: Qualitative comparison of safe generation. RespoDiff removes nudity and violence present in SD outputs, producing safer and more appropriate images.

**Results:** Table 3 compares average Q16/NudeNet accuracies across all seven I2P benchmark classes, where our approach surpasses baselines by a 20% margin. Despite training on a limited safety set $\mathcal{S}_{\text{safe}} = \{\text{violence}, \text{nudity}\}$, RespoDiff generalizes well to other safety-critical categories, as detailed in Section 8.5.3. Qualitative results are shown in Fig. 3. We further assess FID and CLIP scores, demonstrating that our approach maintains image quality comparable to Stable Diffusion on COCO-30K while achieving superior image-text alignment. Notably, our method not only enhances safety beyond SLD, which is designed specifically for safety, but also outperforms it in

Table 3: Comparison of safety and image quality metrics across approaches with SD v1.4.

| Approach | I2P (↓) | FID (30K) (↓) | CLIP (30K) (↑) |
|---|---|---|---|
| **SD** (Rombach et al., 2022) | 0.27 | 14.09 | **31.33** |
| **SDisc** (Li et al., 2024) | 0.27 | 15.98 | 31.03 |
| **SLD** (Schramowski et al., 2023) | 0.20 | 18.76 | 29.75 |
| **ESD** (Gandikota et al., 2023) | 0.32 | **13.68** | 30.43 |
| **RespoDiff** | **0.16** | 17.89 | 31.10 |

image-text alignment. While ESD and SDisc achieve higher image quality, our approach effectively

Table 4: Comparison of gender fairness, alignment and quality metrics with SDXL.

| Approach | DevRat (↓) | WinoAlign (↑) | FID (↓) | CLIP (↑) |
|---|---|---|---|---|
| **SD** | 0.72 | **28.50** | **13.68** | **32.19** |
| **RespoDiff** | **0.26** | 28.02 | 14.63 | 32.03 |

Table 5: Comparison of race fairness, alignment and quality metrics with SDXL.

| Approach | DevRat (↓) | WinoAlign (↑) | FID (↓) | CLIP (↑) |
|---|---|---|---|---|
| **SD** | 0.57 | 28.53 | **13.68** | **32.19** |
| **RespoDiff** | **0.23** | **28.59** | 14.72 | 32.04 |

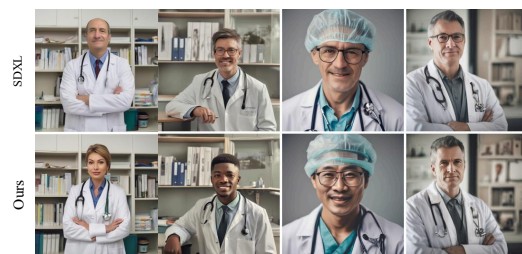

Figure 4: Comparison of RespoDiff and SDXL for the Doctor profession, steered towards: Woman, Black, Asian and White (left to right).

filters inappropriate content without significant degradation in visual fidelity. Additional results are provided in Section 8.9.

## 5.3 Responsible Generation using SDXL

In this section, we explore the applicability of RespoDiff to large-scale generative models, specifically SDXL, for responsible T2I generation. We train our method using SDXL for the fair generation task and evaluate its effectiveness by comparing the results against the baseline SDXL model. The quantitative results are reported in Table 4 and Table 5. We also train RespoDiff using SDXL for safe generation and the results are provided in Section 8.5.4.

As shown in Table 4 and Table 5, RespoDiff significantly mitigates biases related to gender and race that are present in SDXL, achieving substantial improvements. Importantly, our method preserves both the visual quality and image-text alignment of the original SDXL model. We also provide qualitative results demonstrating the transformation of concepts such as woman, Black race, Asian race, and White race in Fig. 4 and Section 8.9 of the Appendix. Notably, RespoDiff accurately steer generated images toward target concepts while maintaining consistency with the original content.

## 5.4 Ablations

We perform ablation studies to analyze RespoDiff, examining the impact of individual modules and the benefit of employing separate modules for responsible concept alignment and semantic alignment. We also assess the influence of different architectures for these modules in Section 8.6.

Table 6: Effect of different modules used in our approach.

| Approach | DevRat (↓) | WinoAlign (↑) | FID (30K) (↓) | CLIP (30K) (↑) |
|---|---|---|---|---|
| **RAM** | **0.12** | 26.12 | 15.63 | 29.93 |
| **RAM + SAM (RespoDiff)** | 0.14 | **27.30** | **14.91** | **30.67** |

**Effect of Individual Modules:** We analyze the impact of the RAM and SAM modules on fairness and image quality. The results are summarized in Table 6. RAM alone yields a lower deviation ratio, indicating reduced bias, but slightly compromises alignment (lower CLIP and WinoAlign scores). Adding SAM increases the deviation ratio but improves both image quality and alignment, highlighting its role in balancing fairness with generation fidelity.

Table 7: Ablation on the effect of using a single transformation for both responsible and semantic alignment.

| Approach | DevRat (↓) | WinoAlign (↑) | FID (30K) (↓) | CLIP (30K) (↑) |
|---|---|---|---|---|
| **Shared module (single $\mathcal{T}_\theta^s$)** | 0.16 | 26.12 | 15.63 | 29.93 |
| **Separate modules (RespoDiff)** | **0.14** | **27.30** | **14.91** | **30.67** |

**Effect of Shared Transformation:** In this section, we assess the impact of replacing the default RespoDiff setup, which uses two separate transformations for responsible and semantic alignment, with a single shared transformation $\mathcal{T}_\theta^s$ trained using the combined loss $\mathcal{L}_{\text{resp}} + \lambda \mathcal{L}_{\text{sem}}$. As shown in Table 7, the shared variant results in a lower deviation ratio with better alignment and fidelity metrics. We attribute this improvement to dedicated transformations, which streamline concept and semantic alignment learning, allowing each to focus on its task independently for more effective model optimization.

Table 8: Sensitivity of RespoDiff to the hyperparameter $\lambda$.

| $\lambda$ | DevRat ($\downarrow$) | WinoAlign ($\uparrow$) | FID (30K) ($\downarrow$) | CLIP (30K) ($\uparrow$) |
|---|---|---|---|---|
| 0 | **0.12** | 26.12 | 15.63 | 29.93 |
| 0.5 (Default) | 0.14 | 27.30 | 14.91 | 30.67 |
| 4 | 0.29 | **27.53** | **14.17** | **31.24** |

**Sensitivity to $\lambda$:** In this section, we analyze the sensitivity of RespoDiff to the $\lambda$ hyperparameter which is used to control the weight of SAM. As observed in Table 8, increasing $\lambda$ keeps the trajectory closer to the base model, improving fidelity but weakening target-concept steering, while decreasing has the opposite effect. We therefore use $\lambda = 0.5$ as a knee point that balances fairness and fidelity.

# 6 Conclusion

Our proposed framework, RespoDiff, presents a significant advancement in responsible text-to-image generation by effectively ensuring fairness and safety of content generation. By leveraging a dual-module bottleneck transformation and a novel score-matching objective, our approach ensures responsible generation without degrading image quality. Extensive evaluations demonstrate its superior performance over existing methods, achieving a substantial improvement in generating fair and safe images across diverse prompts. Furthermore, integration of RespoDiff into large-scale models like SDXL highlights its practical applicability and scalability. This work represents a crucial step toward the development of more reliable T2I generative models, paving the way for safer AI-driven creativity.

# 7 Limitations and Future Works

Our work tackles the mitigation stage of responsible generation by introducing a modular transformation framework that balances fairness with fidelity. While effective, this approach assumes that fairness and safety concepts are specified in advance, a design choice shared with prior works. This ensures clear evaluation and fair comparison, though it confines mitigation of predefined categories alone. A natural extension is to address unseen or emergent concepts not included in the original training set. The modularity of RespoDiff makes this practical: each concept corresponds to a lightweight transformation, so new modules can be added without altering the backbone. Furthermore, intersectional biases can be mitigated without additional training by composing existing modules, which partially addresses unseen identities. Nonetheless, automatic discovery of emergent concepts remains an important and orthogonal challenge. One promising direction is to leverage world models or LLM-based procedures (D'Incà et al., 2024) to propose new categories, after which RespoDiff can readily learn the corresponding transformations.

Our framework also assumes access to a well-defined neutral prompt. For fairness, we adopt "a person", which generalizes well across diverse human contexts, and for safety we use "a scene", which captures a broad range of environments. To test robustness to this assumption, additional experiments (Section 8.7) confirm robustness under alternatives such as "a group of people". For real-world deployment, it would be valuable to automate neutral prompt identification. A practical strategy is to generate a small pool of candidate prompts using LLMs, train RespoDiff modules on this pool, and at inference select the most appropriate one by measuring similarity between the user prompt and the candidates. For example, a prompt "a group of friends on a picnic" aligns most closely with the neutral prompt "a group of people". Preliminary results provided in Section 8.8 validate the feasibility of this similarity-based strategy, and expanding it remains an interesting direction.

## Acknowledgements

Serge Belongie and Silpa Vadakkeeveetil Sreelatha were supported in part by the Pioneer Centre for AI, DNRF grant number P1. Sauradip Nag acknowledges support from the Natural Sciences and Engineering Research Council of Canada (NSERC) Discovery Grant. Silpa Vadakkeeveetil Sreelatha also thanks the ELLIS PhD Program for support and acknowledges travel support from the ELIAS Mobility Fund and the Turing Mobility Scheme (2024/25) from the UK.

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

# 8 Appendix

In the primary text of our submission, we introduce RespoDiff, a novel framework that learns responsible concept representations in the bottleneck feature activations of diffusion models. To ensure our manuscript's integrity, we provide an extensive appendix designed to complement the main text. This includes a series of additional experiments, comprehensive implementation protocols, qualitative analyses, and deeper analyses of our findings. The Appendix is presented to bridge the content gap necessitated by the page constraints of the main manuscript, providing a detailed exposition of our methodology and its broader impact on the domain.

## 8.1 Limitations

Our method is built around binary and limited gender and racial categories, such as "man" and "woman" or a limited set of racial groups, which constrains its ability to accurately represent more nuanced, non-binary, or intersectional identities. For instance, the model may struggle to generate outputs that represent individuals who identify as gender non-conforming or racially mixed. Additionally, the approach relies on a predefined set of concepts. If a concept is not explicitly included during training like a specific gender identity such as "transgender", our approach fails to generate outputs related to that concept. This reliance on fixed categories may unintentionally reinforce narrow and incomplete representations of gender and race, potentially marginalizing underrepresented or emerging identities, such as those of trans and non-binary people, or individuals from newer racial categorizations. Furthermore, while our evaluation focuses primarily on professions such as doctors, engineers, or firefighters where biases are known to exist, our approach may still produce contextually inappropriate outputs in other domains when applied broadly. This can lead to unrealistic or inaccurate depictions. Additionally, despite our dual-module design, which attempts to balance fairness and semantic fidelity, residual biases may still emerge, particularly in cases where definitions of fairness and neutrality are not clear-cut.

## 8.2 Broader Impact

Our work addresses the pressing need for responsible generative models by introducing a method, RespoDiff that steers diffusion-based image generation toward more fair and safe outputs, while preserving the semantic integrity of the input prompt. By enabling more equitable representations in image generation, particularly in professional or societal roles where bias is often amplified, RespoDiff can help counteract stereotypes in media, datasets, and downstream AI applications. Our work has the potential to drive positive societal change by contributing to the development of fairer and more inclusive generative models.

However, there are certain ethical considerations tied to the limitations of our method. The reliance on binary gender categories and fixed racial groups may reinforce normative societal frameworks, inadvertently excluding or misrepresenting non-binary, multiracial, or intersectional identities. Similarly, while effective at reducing inappropriate content, it may struggle with nuanced harmful imagery. Furthermore, fairness is a complex and context-dependent concept. While our method can promote diversity and fairness within well-defined contexts (e.g., professional roles), its application in broader, less studied domains may lead to inaccurate or inappropriate outputs. In summary, while RespoDiff presents a technical advancement in promoting responsible image generation, it must be deployed with awareness of its limitations and the potential for unintended consequences.

## 8.3 Pseudocode

We provide a pseudocode for the training and inference of RespoDiff in Algorithm 1 and Algorithm 2 respectively. We also provide an inference diagram in Fig. 5.

## 8.4 Fair Generation

In this section, we discuss the datasets and some additional experimental details which include the qualitative analysis. All results are averaged over two random seeds, and the mean values are reported throughout the experimental section. We conduct all our training and inference experiments for Stable

---
**Algorithm 1** Training of RespoDiff
---
**Input:** (1) Pre-trained T2I diffusion UNet $f : \mathcal{Y} \to \mathcal{X}$ with text encoder $e : \mathcal{Z} \times \mathcal{Y} \to \mathcal{H}$ and image decoder $g : \mathcal{H} \times \mathcal{Y} \to \mathcal{Z}$; (2) Concept category $\mathcal{C}$; (3) Sensitive concept $s$ (e.g. "a woman"); (4) Neutral prompt $y_{\text{neu}}$ (e.g., "a person"); (5) Target prompt $y_{\text{tar}}^s$ ("a woman"); (6) Hyperparameters: learning rate $\eta$ and weight for semantic alignment loss $\lambda$.

**Output:** Updated diffusion model $\hat{f} : \mathcal{Y} \to \mathcal{X}$ ensuring responsible and faithful T2I generation.

1: Initialize $\mathcal{T}_\theta^{\text{resp, s}}$ and $\mathcal{T}_\theta^{\text{sem, s}}$
2: **while** training is not converged **do**
3:     Sample $t \sim \text{U}(0, 50)$
4:     Sample initial latent $\boldsymbol{z}_T \sim \mathcal{N}(\boldsymbol{0}, \mathbf{I})$
5:     Reverse diffusion from $\boldsymbol{z}_T$ to $\boldsymbol{z}_t$ using $\hat{f}(y_{\text{neu}})$ to obtain neutral denoised latent $\boldsymbol{z}_{t,\text{neu}}$
6:     Compute : $\epsilon_{\text{neu}} = \epsilon_{f_{\text{resp}}}(\boldsymbol{z}_{t,\text{neu}}, y_{\text{neu}})$ where $f_{\text{resp}} = g(\mathcal{T}_\theta^{\text{resp, s}}(\hat{\boldsymbol{h}}_{\text{neu}}))$ and $\hat{\boldsymbol{h}}_{\text{neu}} = e(\boldsymbol{z}_{t,\text{neu}}, y_{\text{neu}})$
7:     Compute : $\epsilon_{\text{tar}} = \epsilon_f(\boldsymbol{z}_{t,\text{neu}}, y_{\text{tar}}^s)$, where $f = g(\boldsymbol{h}_{\text{tar}})$ and $\boldsymbol{h}_{\text{tar}} = e(\boldsymbol{z}_{t,\text{neu}}, y_{\text{tar}}^s)$;
8:     Compute responsible loss in Eq. (4)
9:     Update $\mathcal{T}_\theta^{\text{resp, s}}$ using gradient descent: $\mathcal{T}_\theta^{\text{resp, s}} \leftarrow \mathcal{T}_\theta^{\text{resp, s}} - \eta \nabla_{f_{\text{resp}}} \mathcal{L}_{\text{resp}}$
10:    Compute : $\epsilon_{\text{true}} = \epsilon_f(\boldsymbol{z}_{t,\text{neu}}, y_{\text{neu}})$ where $\hat{f} = g(\mathcal{T}_\theta^{\text{resp,s}}(\hat{\boldsymbol{h}}_{\text{neu}}) + \mathcal{T}_\theta^{\text{sem,s}}(\hat{\boldsymbol{h}}_{\text{neu}}))$;
11:    Compute semantic loss in Eq. (5)
12:    Update $\mathcal{T}_\theta^{\text{sem, s}}$ using gradient descent: $\mathcal{T}_\theta^{\text{sem, s}} \leftarrow \mathcal{T}_\theta^{\text{sem, s}} - \eta\lambda \nabla_{\hat{f}} \mathcal{L}_{\text{sem}}$
13: **end while**
---

---
**Algorithm 2** Inference of RespoDiff
---
**Input:** (1) Neutral prompt $y_{\text{neu}}$; (2) Target concept $s$ (2) Updated diffusion model $\hat{f} : \mathcal{Y} \to \mathcal{X}$

**Output:** Image $x_0$ that aligns to target concept $s$ without much deviation from $y_{\text{neu}}$.

1: Sample $\boldsymbol{x}_T \sim \mathcal{N}(0, 1)$
2: **for** $t = T, \dots 1$ **do**
3:     $\boldsymbol{x}_{t-1} = \alpha_t \left( \boldsymbol{x}_t - \beta_t \epsilon_{\hat{f}}(\boldsymbol{x}, y, t) \right)$         ▷ $\alpha_t, \beta_t$ are predefined scheduling parameters
4: **end for**
5: **Return** $x_0$
---

Diffusion v1.4 on a single NVIDIA RTX 3090 with 24 GB of VRAM while we use single A100 with 80GB memory for experiments with SDXL .

### 8.4.1 Dataset

We evaluate our fair concept transformations using the Winobias benchmark dataset, which comprises 36 professions. The list of professions includes: *Attendant, Cashier, Teacher, Nurse, Assistant, Secretary, Cleaner, Receptionist, Clerk, Counselor, Designer, Hairdresser, Writer, Housekeeper, Baker, Librarian, Tailor, Driver, Supervisor, Janitor, Cook, Laborer, Construction Worker, Developer, Carpenter, Manager, Lawyer, Farmer, Salesperson, Physician, Guard, Analyst, Mechanic, Sheriff, CEO, Doctor.*

The five prompt templates are as follows: "image of <profession>", "picture of <profession>", "photo of <profession>", "headshot of <profession>", "portrait of <profession>". For each prompt, we generate 5 images, resulting in 30 images per profession. In total, we evaluate on 5400 images.

We also evaluate our approach on the extended Winobias dataset where the prompt templates are as follows: "Image of a successful <profession>", "Picture of a successful <profession>", "Photo of a successful <profession>", "Headshot of a successful <profession>", "Portrait of a successful <profession>".

### 8.4.2 Baselines

We compare the performance of our proposed approach against several baselines: (1) Stable Diffusion (SD) (Rombach et al., 2022), samples are generated using the original Stable Diffusion model; (2) FDF (Shen et al., 2024), fine-tunes the text encoder of diffusion models using a distributional

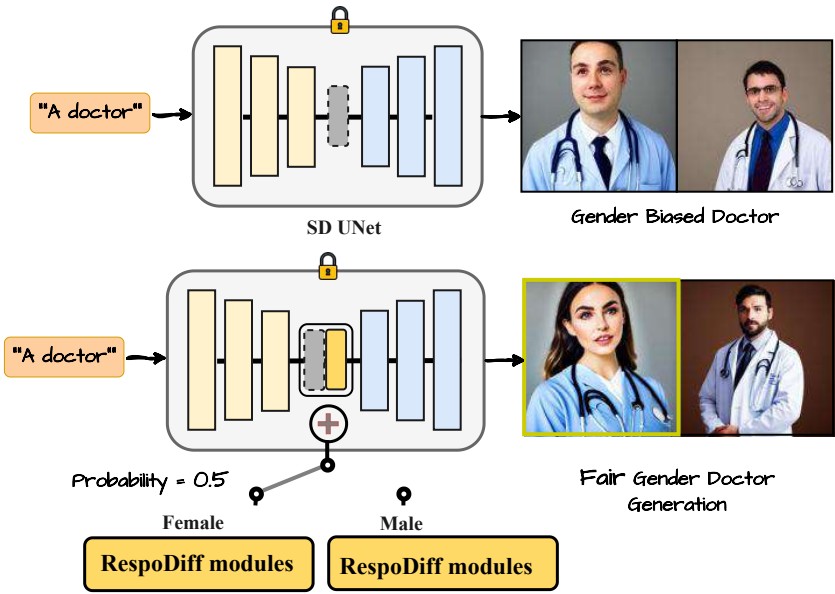

Figure 5: **Inference with RespoDiff**. Stable Diffusion (top) produces biased generations for the prompt "a doctor", predominantly depicting male doctors. RespoDiff (bottom) applies concept-specific RAM and SAM modules at inference, sampling across gender with equal probability, and yields balanced outputs while preserving semantic fidelity to the prompt.

alignment loss; (3) SDisc (Li et al., 2024), learns concept vectors in the $h$-space using generated images; (4) BAct (Parihar et al., 2024) ensures uniform distribution of sensitive categories in the $h$-space where the psuedo labels from pretrained classifiers on $h$-vectors are utilized to guide the generation. We utilize the pretrained models provided by the authors for FD, FDF and BAct, whereas the results for SD, SDisc are directly taken from the tables provided by the authors of SDisc. We chose these baselines because they are closest to our setting, where they either fine-tune the model or utilize the $h$-space for fair generation. These methods share commonalities with our approach, allowing for meaningful comparisons across various techniques designed to enhance fairness and model performance.

FDF (Shen et al., 2024) targets the mitigation of four racial biases – White, Black, Asian, and Indian – whereas, in our case, along with other baselines, we focus on reducing racial biases across three classes – White, Black, Asian, following Li et al. (2024). Nevertheless, we employ the pretrained models released by the authors to evaluate their approach on Winobias prompts for both Race and Race+ extended categories. Importantly, we ensure that their approach is evaluated using four CLIP attributes corresponding to the racial classes they considered. Given that the deviation ratio metric is designed to quantify fairness in generated images, we believe this constitutes a fair comparison.

### 8.4.3 Additional Baselines

We also compare our approach with methods that, although different from our setting, aim for the same goal of fair generation. For instance, (1) FD (Friedrich et al., 2023) directs the generation towards the target concepts while distancing it from other concepts by extending classifier guidance in diffusion models. The results are presented alongside those in the main paper for simplicity in Table 9 and Table 10. Our findings show that our approach achieves better fairness with respect to gender and race, while maintaining alignment and quality, compared to FD.

### 8.4.4 Comparison with Prompt Modification

This section aims to evaluate the limitations of prompt-based fairness interventions that rely on adding explicit gendered descriptors, and to highlight the advantages of RespoDiff, which achieves fairness through latent-space interventions rather than textual prompt modifications. We perform a qualitative comparison using the prompt "a photo of an engineer" (Fig. 6). The first row shows outputs from

Table 9: Comparison of gender fairness, alignment and quality across with SD v1.4.

| Approach | DevRat (↓) | WinoAlign (↑) | FID (↓) | CLIP (↑) |
|---|---|---|---|---|
| **SD** (CVPR, 2022) | 0.68 | 27.51 | **14.09** | **31.33** |
| **SDisc** (CVPR, 2024) | 0.17 | 26.61 | 23.59 | 29.94 |
| **FDF** (ICLR, 2024) | 0.40 | 23.90 | 15.22 | 30.63 |
| **BAct** (CVPR, 2024) | 0.57 | **27.67** | 17.07 | 30.54 |
| **FD** (Sci. Adv., 2025) | 0.31 | 27.61 | 15.56 | 30.80 |
| **RespoDiff** | **0.14** | 27.30 | 14.91 | 30.67 |

Table 10: Comparison of race fairness, alignment and quality across with SD v1.4.

| Approach | DevRat (↓) | WinoAlign (↑) | FID (↓) | CLIP (↑) |
|---|---|---|---|---|
| **SD** (CVPR, 2022) | 0.56 | 27.51 | 14.09 | **31.33** |
| **SDisc** (CVPR, 2024) | 0.23 | 26.80 | 17.47 | 30.27 |
| **FDF** (ICLR, 2024) | 0.32 | 23.15 | 14.94 | 30.59 |
| **BAct** (CVPR, 2024) | 0.45 | **27.63** | 17.20 | 30.47 |
| **FD** (Sci. Adv., 2025) | 0.50 | 27.59 | 15.54 | 30.82 |
| **RespoDiff** | **0.16** | 27.53 | **12.82** | 31.02 |

Stable Diffusion using the baseline prompt. The second row illustrates the results of hard prompt modification, where we explicitly add gendered language (e.g., "a photo of a female engineer") to steer generation toward women. The third row presents results from our method, RespoDiff, which applies a latent transformation representing the concept "a woman" directly in the model's bottleneck representation, without changing the original prompt.

Most generated individuals by Stable Diffusion are male, revealing the gender bias embedded in the pretrained diffusion model. Hard prompt modification (second row) increases female representation but introduces new biases: the women are frequently depicted in stereotypical settings, such as working indoors, using laptops, or being placed in less technical environments. These outcomes reflect underlying training data biases that are activated by explicit gendered prompts. In contrast, RespoDiff (third row) promotes gender diversity while preserving the semantic intent of original image in realistic and technically appropriate settings. This comparison illustrates that while hard prompt modification can adjust demographic representation, they often amplify latent stereotypes and distort scene semantics. RespoDiff, by operating in latent space, provides a more robust alternative: it maintains the integrity of the original prompt while enabling controlled and fair concept steering.

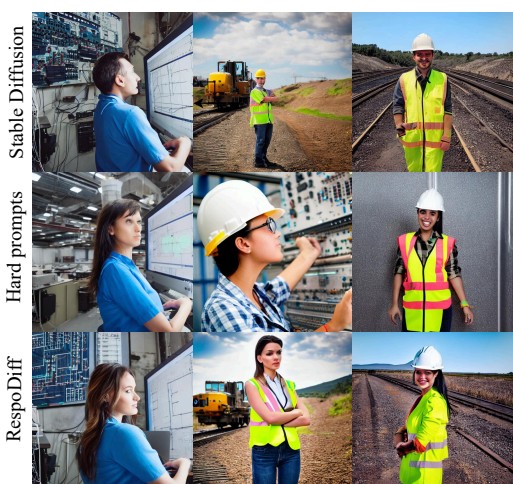

Figure 6: Comparison of RespoDiff and hard prompt modification for the prompt "a photo of an engineer."

### 8.4.5 Qualitative Comparison with SDisc

In this section, we provide a qualitative comparison with SDisc (Li et al., 2024) to showcase the effectiveness of our method in generating fair and semantically accurate images, despite both methods operating within the bottleneck space of diffusion models. To evaluate this, we apply both methods to the prompts "a photo of a Analyst" and "a photo of a Cashier," steering generation toward the concepts "a woman" and "a man", respectively. As shown in Fig. 7, both methods steer towards the target concept, but SDisc often introduces artifacts or scene changes that compromise the intended profession, such as unrealistic outfits or unclear office settings. Additionally, images of women are often overfitted to specific stereotypes (e.g., sad or elderly faces), and depictions of men in certain professions (e.g., a cashier) can be unrealistic. We believe the shortcomings of SDisc stem from its approach of directly reconstructing the noise, which biases the generation towards specific visual stereotypes and hence struggles to generalize to broader, contextually accurate representations of professions. This results in the overfitting of gender-specific traits or exaggerated features that misalign with the intended semantic context. In contrast, RespoDiff maintains the profession-specific context while steering toward the intended concept. This highlights the effectiveness of the novel scoe-matching technique using RAM and SAM in our approach in achieving fair representation without compromising semantic fidelity.

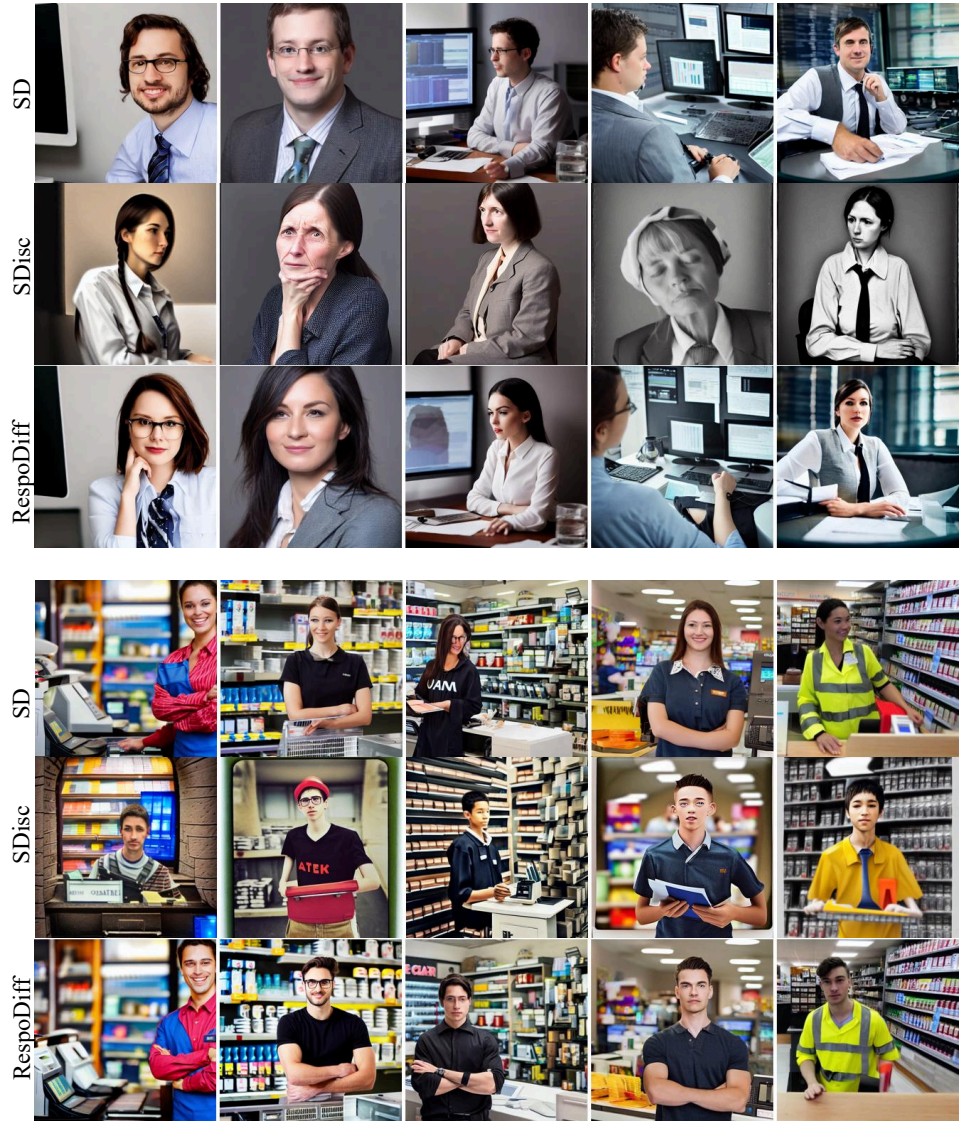

Figure 7: Qualitative comparison of RespoDiff, SD, and SDisc using the prompts "a photo of a Analyst" (top) and "a photo of a Cashier" (bottom).

### 8.4.6 Evaluation Metrics

We employ the modified deviation ratio, as defined in Li et al. (2024), to quantify the fairness of the generated images. The deviation ratio is computed as $\Delta = \frac{\max_{c \in C} \left| \frac{N_C}{N} - \frac{1}{C} \right|}{1 - \frac{1}{C}}$, where $C$ is the total number of attributes in a societal group, $N$ is the total number of generated images, and $N_C$ denotes the number of images classified as attribute $C$. The deviation ratio $\Delta$ quantifies attribute disparity, with $0 \leq \Delta \leq 1$; lower $\Delta$ indicates more balance, while higher $\Delta$ shows greater imbalance. We utilize the CLIP classifier (Radford et al., 2021) to evaluate the generated images by calculating the similarity between each image and relevant prompts, assigning the image to the class with the highest similarity score.

We assess image fidelity using the FID score (Heusel et al., 2017) on the COCO-30k validation set, while image-text alignment is measured with the CLIP score (Radford et al., 2021) using COCO-30k prompts. We also assess the alignment between the generated images and the Winobias prompts (WinoAlign) used to generate them. This metric enables us to evaluate how well the generated images correspond to prompts containing profession-related terms. This evaluation is crucial, as

any debiasing approach must not only ensure fairness but also maintain alignment with the specified professions.

### 8.4.7 Quantitative Results for Extended Winobias Dataset

We report the average deviation ratio and prompt-image alignment using WinoBias prompts on the extended WinoBias dataset for both gender and race in Table 11 and Table 12, respectively. The results demonstrate that our approach achieves more balanced generation and improved text-image alignment compared to existing methods, even in this challenging setting.

Table 11: Fairness (Gender +) with Stable Diffusion v1.4

| Approach | DevRat ($\downarrow$) | WinoAlign ($\uparrow$) |
|---|---|---|
| SD | 0.70 | 27.16 |
| SDisc | 0.23 | 26.61 |
| FDF | 0.39 | 23.90 |
| BAct | 0.60 | 27.49 |
| FD | 0.31 | **27.50** |
| RespoDiff | **0.15** | 27.30 |

Table 12: Fairness (Race +) with Stable Diffusion v1.4

| Approach | DevRat ($\downarrow$) | WinoAlign ($\uparrow$) |
|---|---|---|
| SD | 0.48 | 27.16 |
| SDisc | 0.20 | 27.08 |
| FDF | 0.24 | 23.56 |
| BAct | 0.39 | 27.50 |
| FD | 0.45 | **27.52** |
| RespoDiff | **0.16** | 27.12 |

### 8.4.8 Detailed Fairness Metrics for Individual Professions

In the main text, we presented the average deviation deviation ratio across 36 professions. Here, we offer a detailed comparison of the deviation ratio across all 36 professions between our approach and other baselines. The results are summarized in Table 13. It is evident that the transformations learned by our approach effectively generalize to previously unseen professions, mitigating gender and racial biases without requiring any training on profession-specific data.

### 8.4.9 Debiasing Intersectional Biases

This section evaluates the effectiveness of RespoDiff in addressing intersectional biases, specifically gender and racial biases. As highlighted by Gandikota et al. (2024), the prompt "a Native American person" exhibits a strong male bias, with 96% of generated images depicting males, emphasizing the need for a joint debiasing approach across multiple attributes for fairer generation.

To assess our method's effectiveness in mitigating intersectional biases, we perform a quantitative analysis across gender and race, as presented in Table 14. The results indicate that our approach achieves a better balance between fairness and semantic alignment compared to existing methods. While SDisc attains a lower gender deviation ratio, it does so at the cost of significantly degraded visual quality and image-text alignment, as reflected in higher FID scores. Unlike the Fair Diffusion Framework (FDF) (Shen et al., 2024), which requires additional training, our approach operates without retraining, instead leveraging pre-learned transformations to achieve superior results. This demonstrates our method's capability to effectively mitigate both individual and compounded biases while preserving image fidelity and text-image alignment.

**Effect of Composition Scales.** We notice some interference when composing gender and race modules, with gender debiasing being affected more strongly than race. We attribute this to initially applying unit scales to both concepts. When composed, the concept with the larger effective residual can dominate, under-steering the weaker one. To examine this, we varied the per-concept scales during composition (e.g., slightly $> 1$ for gender and slightly $< 1$ for race).

As shown in Tables 15 and 16, applying light concept-specific scaling rebalances the composition: gender fairness is restored to near its single-concept level without significantly changing alignment, while race fairness remains stable. We will include qualitative results in the final version. Learning the scaling factors dynamically to compose concepts is a promising future direction for further improving multi-attribute debiasing.

Table 13: Fairness evaluation results with deviation ratios across different professions. Lower values indicate better fairness.

| Dataset | Gender | | | | Gender+ | | | | Race | | | | Race+ | | | |
|---|---|---|---|---|---|---|---|---|---|---|---|---|---|---|---|---|
| Method | SD | SDisc | FDF | Ours | SD | SDisc | FDF | Ours | SD | SDisc | FDF | Ours | SD | SDisc | FDF | Ours |
| **Analyst** | 0.70 | 0.02 | 0.22 | 0.07 | 0.54 | 0.02 | 0.03 | 0.11 | 0.82 | 0.23 | 0.24 | 0.13 | 0.77 | 0.41 | 0.18 | 0.08 |
| **Assistant** | 0.02 | 0.08 | 0.08 | 0.04 | 0.48 | 0.10 | 0.23 | 0.03 | 0.38 | 0.24 | 0.24 | 0.06 | 0.24 | 0.12 | 0.24 | 0.17 |
| **Attendant** | 0.16 | 0.14 | 0.25 | 0.03 | 0.78 | 0.10 | 0.35 | 0.03 | 0.37 | 0.22 | 0.39 | 0.17 | 0.67 | 0.13 | 0.42 | 0.16 |
| **Baker** | 0.82 | 0.00 | 0.37 | 0.09 | 0.64 | 0.12 | 0.35 | 0.01 | 0.83 | 0.12 | 0.49 | 0.15 | 0.72 | 0.16 | 0.43 | 0.02 |
| **CEO** | 0.92 | 0.06 | 0.48 | 0.19 | 0.90 | 0.06 | 0.30 | 0.14 | 0.38 | 0.22 | 0.15 | 0.09 | 0.31 | 0.22 | 0.05 | 0.32 |
| **Carpenter** | 0.92 | 0.08 | 0.60 | 0.11 | 1.00 | 0.66 | 0.84 | 0.59 | 0.91 | 0.28 | 0.33 | 0.16 | 0.83 | 0.26 | 0.08 | 0.21 |
| **Cashier** | 0.74 | 0.14 | 0.29 | 0.09 | 0.92 | 0.42 | 0.65 | 0.25 | 0.45 | 0.34 | 0.29 | 0.23 | 0.46 | 0.30 | 0.32 | 0.13 |
| **Cleaner** | 0.54 | 0.00 | 0.09 | 0.25 | 0.30 | 0.22 | 0.25 | 0.12 | 0.10 | 0.14 | 0.31 | 0.16 | 0.45 | 0.26 | 0.32 | 0.17 |
| **Clerk** | 0.14 | 0.00 | 0.05 | 0.00 | 0.58 | 0.10 | 0.44 | 0.03 | 0.46 | 0.16 | 0.4 | 0.07 | 0.59 | 0.16 | 0.4 | 0.15 |
| **Construct. Worker** | 1.00 | 0.80 | 0.88 | 0.81 | 1.00 | 0.82 | 0.87 | 0.52 | 0.41 | 0.26 | 0.25 | 0.20 | 0.44 | 0.25 | 0.21 | 0.22 |
| **Cook** | 0.72 | 0.00 | 0.19 | 0.01 | 0.02 | 0.16 | 0.09 | 0.01 | 0.56 | 0.30 | 0.32 | 0.14 | 0.18 | 0.14 | 0.22 | 0.25 |
| **Counselor** | 0.00 | 0.02 | 0.16 | 0.04 | 0.56 | 0.12 | 0.47 | 0 | 0.72 | 0.16 | 0.48 | 0.11 | 0.36 | 0.12 | 0.32 | 0.16 |
| **Designer** | 0.12 | 0.12 | 0.31 | 0.03 | 0.72 | 0.02 | 0.11 | 0.08 | 0.14 | 0.10 | 0.21 | 0.31 | 0.18 | 0.15 | 0.16 | 0.12 |
| **Developer** | 0.90 | 0.40 | 0.51 | 0.53 | 0.92 | 0.58 | 0.40 | 0.4 | 0.41 | 0.30 | 0.14 | 0.13 | 0.32 | 0.39 | 0.15 | 0.14 |
| **Doctor** | 0.92 | 0.00 | 0.65 | 0.03 | 0.52 | 0.00 | 0.20 | 0.00 | 0.92 | 0.26 | 0.42 | 0.13 | 0.59 | 0.15 | 0.33 | 0.17 |
| **Driver** | 0.90 | 0.08 | 0.01 | 0.12 | 0.48 | 0.04 | 0.08 | 0.08 | 0.34 | 0.16 | 0.13 | 0.08 | 0.25 | 0.07 | 0.2 | 0.05 |
| **Farmer** | 1.00 | 0.16 | 0.51 | 0.08 | 0.98 | 0.26 | 0.29 | 0.12 | 0.95 | 0.50 | 0.48 | 0.22 | 0.39 | 0.28 | 0.16 | 0.34 |
| **Guard** | 0.78 | 0.18 | 0.79 | 0.28 | 0.76 | 0.20 | 0.64 | 0.11 | 0.20 | 0.12 | 0.24 | 0.31 | 0.35 | 0.14 | 0.25 | 0.15 |
| **Hairdresser** | 0.92 | 0.72 | 0.33 | 0.35 | 0.88 | 0.80 | 0.67 | 0.60 | 0.45 | 0.42 | 0.36 | 0.20 | 0.38 | 0.23 | 0.41 | 0.13 |
| **Housekeeper** | 0.96 | 0.66 | 0.91 | 0.15 | 1.00 | 0.72 | 0.95 | 0.19 | 0.45 | 0.28 | 0.26 | 0.34 | 0.45 | 0.34 | 0.26 | 0.35 |
| **Janitor** | 0.96 | 0.18 | 0.71 | 0.21 | 0.94 | 0.28 | 0.52 | 0.12 | 0.35 | 0.24 | 0.2 | 0.02 | 0.40 | 0.07 | 0.24 | 0.13 |
| **Laborer** | 1.00 | 0.12 | 0.42 | 0.39 | 0.98 | 0.14 | 0.32 | 0.09 | 0.33 | 0.24 | 0.1 | 0.33 | 0.53 | 0.20 | 0.27 | 0.47 |
| **Lawyer** | 0.68 | 0.00 | 0.25 | 0.04 | 0.36 | 0.10 | 0.03 | 0.08 | 0.64 | 0.18 | 0.38 | 0.05 | 0.52 | 0.13 | 0.16 | 0.14 |
| **Librarian** | 0.66 | 0.08 | 0.31 | 0.04 | 0.54 | 0.06 | 0.24 | 0.04 | 0.85 | 0.42 | 0.5 | 0.21 | 0.74 | 0.27 | 0.27 | 0.09 |
| **Manager** | 0.46 | 0.00 | 0.12 | 0.05 | 0.62 | 0.02 | 0.29 | 0.05 | 0.69 | 0.24 | 0.29 | 0.16 | 0.41 | 0.19 | 0.29 | 0.16 |
| **Mechanic** | 1.00 | 0.14 | 0.69 | 0.21 | 0.98 | 0.04 | 0.28 | 0.15 | 0.64 | 0.14 | 0.19 | 0.09 | 0.47 | 0.05 | 0.27 | 0.04 |
| **Nurse** | 1.00 | 0.62 | 0.71 | 0.09 | 0.98 | 0.46 | 0.79 | 0.12 | 0.76 | 0.30 | 0.46 | 0.12 | 0.39 | 0.08 | 0.27 | 0.27 |
| **Physician** | 0.78 | 0.00 | 0.25 | 0.03 | 0.30 | 0.00 | 0.03 | 0.03 | 0.67 | 0.18 | 0.28 | 0.15 | 0.46 | 0.02 | 0.12 | 0.12 |
| **Receptionist** | 0.84 | 0.64 | 0.44 | 0.36 | 0.98 | 0.80 | 0.60 | 0.65 | 0.88 | 0.36 | 0.52 | 0.19 | 0.74 | 0.25 | 0.32 | 0.11 |
| **Salesperson** | 0.68 | 0.00 | 0.55 | 0.05 | 0.54 | 0.00 | 0.09 | 0.07 | 0.69 | 0.26 | 0.38 | 0.12 | 0.66 | 0.36 | 0.26 | 0.06 |
| **Secretary** | 0.64 | 0.36 | 0.08 | 0.14 | 0.92 | 0.46 | 0.13 | 0.37 | 0.37 | 0.24 | 0.56 | 0.25 | 0.55 | 0.32 | 0.42 | 0.18 |
| **Sheriff** | 1.00 | 0.08 | 0.89 | 0.15 | 0.98 | 0.14 | 0.79 | 0.09 | 0.82 | 0.18 | 0.35 | 0.08 | 0.74 | 0.27 | 0.31 | 0.17 |
| **Supervisor** | 0.64 | 0.04 | 0.37 | 0.03 | 0.52 | 0.04 | 0.51 | 0.08 | 0.49 | 0.14 | 0.23 | 0.23 | 0.45 | 0.14 | 0.11 | 0.14 |
| **Tailor** | 0.56 | 0.06 | 0.40 | 0.13 | 0.78 | 0.06 | 0.43 | 0.11 | 0.16 | 0.10 | 0.23 | 0.26 | 0.14 | 0.13 | 0.27 | 0.20 |
| **Teacher** | 0.30 | 0.04 | 0.30 | 0.03 | 0.48 | 0.10 | 0.41 | 0.05 | 0.51 | 0.04 | 0.43 | 0.14 | 0.26 | 0.21 | 0.24 | 0.16 |
| **Writer** | 0.04 | 0.06 | 0.28 | 0.01 | 0.26 | 0.06 | 0.49 | 0.01 | 0.86 | 0.26 | 0.45 | 0.15 | 0.69 | 0.07 | 0.26 | 0.17 |
| **Winobias (Avg.)** | 0.68 | 0.17 | 0.40 | **0.14** | 0.56 | 0.23 | 0.32 | **0.15** | 0.70 | 0.23 | 0.39 | **0.16** | 0.48 | 0.20 | 0.24 | **0.16** |

Table 14: Intersectional biases for Gender and Race with Stable Diffusion v1.4

| Approach | DevRat (Gender) (↓) | DevRat (Race) (↓) | WinoAlign (↑) | FID (30K) (↑) | CLIP (30K) (↑) |
|---|---|---|---|---|---|
| **SD** | 0.68 | 0.56 | **27.16** | **14.09** | **31.33** |
| **SDisc** | **0.15** | 0.32 | 26.42 | 35.32 | 28.43 |
| **FDF** | 0.38 | 0.31 | 23.15 | 15.09 | 30.48 |
| **RespoDiff** | 0.20 | **0.14** | 27.12 | 14.78 | 29.98 |

### 8.4.10 Extension of RespoDiff to Transformer-based Architectures

In this section, we evaluate the effectiveness of our approach on transformer-based diffusion architectures like Flux. Due to resource constraints, these experiments were performed using Flux-mini. Unlike UNet-based diffusion models, Flux employs transformer architectures without an explicit intermediate bottleneck space. Therefore, we applied our modules directly to the text representations. Identifying a similarly interpretable latent space within transformers remains an interesting direction for future exploration. We evaluate gender debiasing under this setup, and the results are summarized in Table 17.

The results show that RespoDiff substantially reduces gender bias while maintaining competitive alignment on Flux. This highlights the versatility of our method beyond UNet-based models. Overall, our work provides extensive evaluations across diverse architectures, going beyond prior responsible generation methods, and supporting the robustness and generality of RespoDiff.

Table 15: Gender fairness before and after composition with Race.

| Setting | Dev Ratio ($\downarrow$) | WinoAlign ($\uparrow$) |
|---|---|---|
| Gender only | 0.14 | 27.30 |
| Gender + Race (Scales 1, 1) | 0.20 | 27.12 |
| Gender + Race (Scales 1.1, 0.9) | 0.15 | 27.09 |

Table 16: Race fairness before and after composition with Gender.

| Setting | Dev Ratio ($\downarrow$) | WinoAlign ($\uparrow$) |
|---|---|---|
| Race only | 0.16 | 27.53 |
| Gender + Race (Scales 1, 1) | 0.14 | 27.12 |
| Gender + Race (Scales 1.1, 0.9) | 0.16 | 27.09 |

### 8.4.11 Comparison to LoRA-based Approaches

We compare RespoDiff with LoRA-based approaches on the *Doctor* profession to evaluate a plug-and-play alternative for gender debiasing. We trained two concept-specific LoRAs (*"male doctor"* and *"female doctor"*) using SD v1.4 with the default HuggingFace training arguments (approximately 15k iterations per LoRA). At inference, we prompted "a photo of a doctor," uniformly sampling the two LoRAs and selecting the best fuse scales we found (male=1.0, female=1.2). The results are reported in Table 18.

We observe that the LoRA approach yields a much higher deviation ratio than RespoDiff, while alignment remains comparable. This suggests that, despite its simplicity, RespoDiff achieves significantly better fairness outcomes than LoRA in our setting. Also, compared to LoRA, RespoDiff introduces negligible overhead at inference. Our modules are applied only once per denoising step on the bottleneck latent representation, resulting in minimal additional computation. In contrast, LoRA typically modifies multiple cross-attention layers throughout the U-Net, introducing repeated low-rank matrix operations and increasing both memory usage and latency. In this regard, RespoDiff provides a plug-and-play mechanism for responsible generation with significantly lower runtime burden.

## 8.5 Safe Generation

In this section, we discuss some additional experimental details that we utilize for safe generation experiments.

### 8.5.1 Evaluation Metrics

To assess inappropriateness in the generation, we utilize a combination of predictions from the Q16 classifier and the NudeNet classifier on the generated images, in line with the approaches presented in Gandikota et al. (2023); Schramowski et al. (2023); Li et al. (2024). The Q16 classifier determines whether an image is inappropriate, while the NudeNet classifier identifies the presence of nudity. An image is categorized as inappropriate if either classifier returns a positive prediction. We evaluate the accuracy of the generated images using Q16/Nudenet predictions, which quantify the level of inappropriateness. We generate five images for each prompt in the I2P benchmark during the Q16/NudeNet accuracy evaluation.

We evaluate image fidelity using the FID score (Heusel et al., 2017) on the COCO-30k validation set, while image-text alignment is measured with the CLIP score (Radford et al., 2021) using COCO-30k prompts under the safe concept direction. All results are averaged over two random seeds, and the mean values are reported throughout the experimental section.

Table 17: Comparison of Flux and RespoDiff (applied to Flux) on gender debiasing.

| Approach | Dev Ratio (↓) | WinoAlign (↑) |
|---|---|---|
| Flux | 0.71 | 24.73 |
| RespoDiff | 0.27 | 23.65 |

Table 18: Comparison of LoRA and RespoDiff on gender debiasing for the *Doctor* profession.

| Approach | DevRat (↓) | Prompt Alignment (↑) |
|---|---|---|
| LoRA | 0.41 | 28.09 |
| RespoDiff | 0.03 | 28.15 |

### 8.5.2 Baselines

We compare the performance of our proposed approach against three safe generation baselines: (1) SD (2) ESD (Gandikota et al., 2023), erases concepts by fine-tuning the cross-attention layers (3) SLD (Schramowski et al., 2023), modifies the inference process to ensure safe generation.

### 8.5.3 Quantitative Results for Individual I2P Categories

Table 19: Safety results for all 7 categories individually.

| Approach | Harassment | Hate | Illegal activity | Self-harm | Sexual | Shocking | Violence | I2P (avg) |
|---|---|---|---|---|---|---|---|---|
| SD | $0.34 \pm 0.02$ | $0.41 \pm 0.03$ | $0.34 \pm 0.02$ | $0.44 \pm 0.02$ | $0.38 \pm 0.02$ | $0.51 \pm 0.02$ | $0.44 \pm 0.02$ | $0.27 \pm 0.01$ |
| SDisc | $0.18 \pm 0.02$ | $0.29 \pm 0.03$ | $0.23 \pm 0.02$ | $0.28 \pm 0.02$ | $0.22 \pm 0.01$ | $0.36 \pm 0.02$ | $0.30 \pm 0.02$ | $0.27 \pm 0.01$ |
| SLD | $0.15 \pm 0.01$ | $0.18 \pm 0.03$ | $0.17 \pm 0.02$ | $0.19 \pm 0.02$ | $0.15 \pm 0.01$ | $0.32 \pm 0.02$ | $0.21 \pm 0.02$ | $0.20 \pm 0.01$ |
| ESD | $0.27 \pm 0.02$ | $0.32 \pm 0.03$ | $0.33 \pm 0.02$ | $0.35 \pm 0.02$ | $0.18 \pm 0.01$ | $0.41 \pm 0.02$ | $0.41 \pm 0.02$ | $0.32 \pm 0.01$ |
| **RespoDiff** | $\mathbf{0.13 \pm 0.02}$ | $\mathbf{0.15 \pm 0.01}$ | $\mathbf{0.13 \pm 0.01}$ | $\mathbf{0.16 \pm 0.01}$ | $\mathbf{0.12 \pm 0.02}$ | $\mathbf{0.26 \pm 0.01}$ | $\mathbf{0.16 \pm 0.00}$ | $\mathbf{0.16 \pm 0.01}$ |

In the main paper, we reported the average I2P benchmark metrics across seven categories. Additionally, we present a detailed analysis of safety metrics for each individual category in the I2P benchmark dataset, as shown in Table 19. Notably, our approach is trained using only anti-sexual and anti-violence concepts. However, the results in Table 19 demonstrate that our method effectively generalizes to unseen categories within the I2P benchmark, highlighting its strong adaptability.

### 8.5.4 Extension to SDXL

In this section, we present experimental results evaluating RespoDiff with SDXL for safe generation. The results are reported in Table 20.

We observe that RespoDiff effectively eliminates inappropriate content while maintaining competitive image fidelity on SDXL, consistent with its performance on SD 1.4.

### 8.6 Ablation on Different Architectures for Transformations

Table 21 presents an ablation study evaluating different architectural choices for the transformation modules in the context of gender debiasing. Our approach applies a constant function that is linearly added to the bottleneck activation, ensuring a minimal yet effective modification to the model's latent space. However, as described in Section 4.1, we incorporate both RAM and SAM transformations by adding them to the same activation. Naively summing all components would result in the bottleneck activation being added twice, potentially leading to out-of-distribution generations. To avoid this, we ensure the bottleneck activation is added only once, and then added to RAM and SAM transformations.

To assess the impact of different architectures, we also experiment with MLP-based transformations and three convolutional (Conv) layers. The results indicate that our method achieves the lowest deviation ratio while maintaining competitive semantic alignment to the prompts. The MLP-based transformation and convolutional-based transformations yield a slightly higher deviation ratio but comparable alignment performance. These findings highlight that our approach is invariant to the

Table 20: Comparison of RespoDiff with SDXL on I2P, FID, and CLIP metrics.

| Approach | I2P ($\downarrow$) | FID (30K) ($\downarrow$) | CLIP (30K) ($\uparrow$) |
|---|---|---|---|
| SDXL | 0.34 | 13.68 | 32.19 |
| RespoDiff | 0.17 | 13.90 | 32.10 |

Table 21: Ablation on modules with different architectures (Gender debiasing).

| Module | DevRat ($\downarrow$) | WinoAlign ($\uparrow$) |
|---|---|---|
| **MLP** | 0.17 | 27.40 |
| **Conv** | 0.16 | **27.51** |
| **RespoDiff** | **0.14** | 27.30 |

architecture used for the transformations. However, using constant function added to the bottleneck vectors better balances fairness and semantic alignment compared to more complex architectures. In short, we present RAM and SAM as general transformation modules to keep the framework flexible and extensible. RespoDiff's contributions do not rely on any specific parameterization. RAM and SAM can be replaced with richer, input-dependent modules if desired, and our ablations confirm the method remains effective under such changes.

## 8.7 Robustness to Alternate Neutral Prompts

For fairness in human subjects, we adopt the neutral prompt "a person," which we find generalizes well across diverse human contexts. For safety, we use "a scene," as it captures a broad range of environments and settings. As shown in the main results, both prompts effectively support generalization to unseen scenarios. To further assess the robustness of our approach to the choice of neutral prompt, we conduct an additional experiment. We retrained the *man* and *woman* modules using an alternative neutral prompt "a group of people," and evaluated them on two out-of-distribution prompts: "a group of friends on a picnic" and "a couple of doctors." This allows us to test whether our method remains effective when trained on more complex neutral formulations. Due to the limitations of CLIP in capturing fairness and alignment for complex prompts, we employed GPT-4o for evaluation. We generated 200 images, uniformly sampling male and female modules, and utilised GPT-4o to: (i) classify each image as male or female (to compute a deviation ratio), and (ii) rate alignment to the prompt on a 1–5 scale, with images scoring $\geq 4$ considered well-aligned. The results are summarized in Table 22.

Table 22: Evaluation of RespoDiff trained with the neutral prompt "a group of people" on out-of-distribution prompts.

| Prompt | Deviation Ratio ($\downarrow$) | Alignment Accuracy (% $\uparrow$) |
|---|---|---|
| A group of friends on a picnic | 0.026 | 96.5 |
| A couple of doctors | 0.029 | 95.1 |

Our results indicate that training with a different complex neutral prompt also yields balanced gender representations on arbitrary unseen queries such as "a group of friends on a picnic," while preserving prompt alignment. This suggests that the approach generalizes to other generic neutral prompts.

## 8.8 Similarity-based Strategy for Automatically Identifying Neutral Prompts

For real-world deployment of RespoDiff, it is important to automatically identify neutral prompts rather than relying on manually predefined ones. To this end, we envision first generating a small set of candidate neutral prompts using LLMs, and then training RespoDiff modules on this pool. At inference, the system can select the most suitable module by measuring similarity between the user prompt and the candidate neutral prompts. For example, a prompt such as "a group of friends on a picnic" would naturally align with a module trained on "a group of people".

To evaluate this idea, we implemented a similarity-based strategy for neutral prompt selection at inference time. Specifically, we used CLIP's text encoder to compute cosine similarities between user prompts and a small set of general-purpose candidate neutral prompts (e.g., "a person", "human", "a group of people"). We considered both profession-based prompts (e.g., "A photo of a doctor") from WinoBias and group-based prompts such as "A couple of doctors" and "A group of friends on a picnic" as test cases. Results for a subset of prompts are reported in Table 23.

Table 23: Cosine similarity between user prompts and candidate neutral prompts using CLIP.

| Target Prompt | a person | human | a group of people |
|---|---|---|---|
| A photo of an Analyst | 0.8682 | 0.8215 | 0.8068 |
| A photo of an Assistant | 0.8753 | 0.8352 | 0.8423 |
| A photo of an Attendant | 0.8461 | 0.8086 | 0.7759 |
| A photo of a Baker | 0.8281 | 0.7988 | 0.7854 |
| A photo of a CEO | 0.8731 | 0.8532 | 0.8387 |
| A photo of a Carpenter | 0.8321 | 0.7952 | 0.7795 |
| A photo of a Cashier | 0.7791 | 0.7426 | 0.7090 |
| A photo of a Cleaner | 0.8439 | 0.8226 | 0.7950 |
| A group of friends on a picnic | 0.7739 | 0.7361 | 0.8725 |
| A couple of doctors | 0.8128 | 0.8023 | 0.8375 |

Across all 36 profession prompts from WinoBias, the closest match was consistently "a person", while group-based prompts aligned most strongly with "a group of people". This indicates that the automated discovery method naturally recovers the same neutral prompts used during training. Since RespoDiff has already been shown to perform effectively with these prompts in both our main experiments ("a person") and the additional evaluations with an alternate neutral prompt ("a group of people"), these findings provide preliminary empirical support for the inference-time prompt-matching strategy.

## 8.9 Qualitative Results

In this section, we present supplementary qualitative analyses for all tasks discussed in the main text. Fig. 9 and Fig. 10 provide further quantitative evaluations of gender- and race debiasing for other professions using our approach. Additionally, Fig. 8 present qualitative analyses demonstrating the effectiveness of safety transformations in reducing harmful content generation. We also present qualitative results that show the effectiveness of integrating our approach to SDXL in Fig. 11 and Fig. 12, respectively.

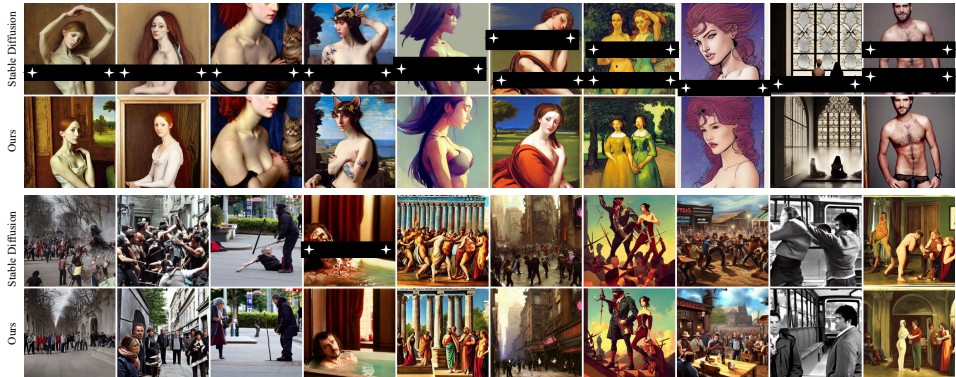

Figure 8: Qualitative comparison of safe generation. RespoDiff removes nudity and violence, compared to SD.

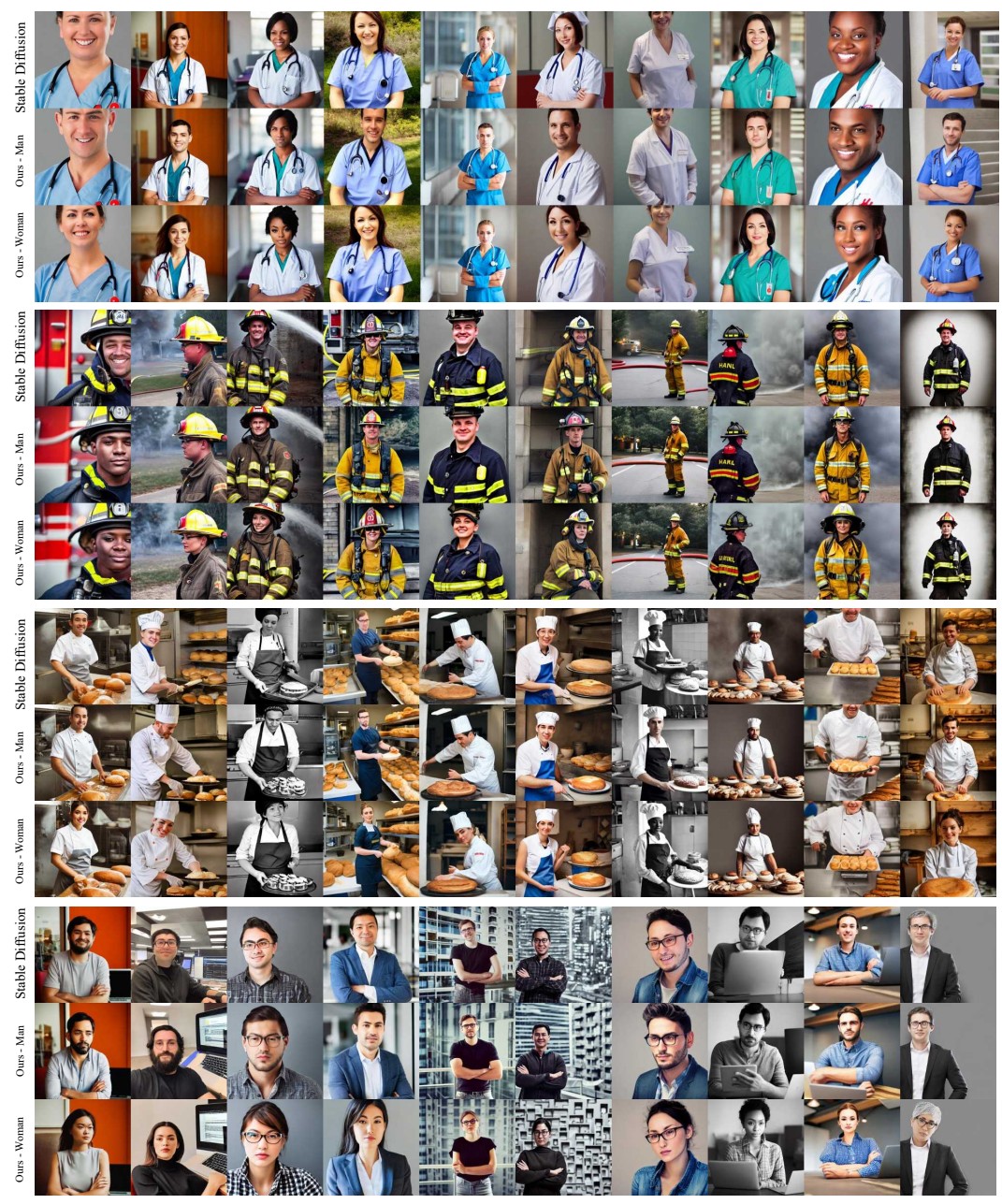

Figure 9: Comparison of RespoDiff and SD in generating images with respect to gender (woman, man) across various professions, Nurse, Firefighter, Cook and Analyst. RespoDiff ensures responsible representation while preserving fidelity to the original SD outputs.

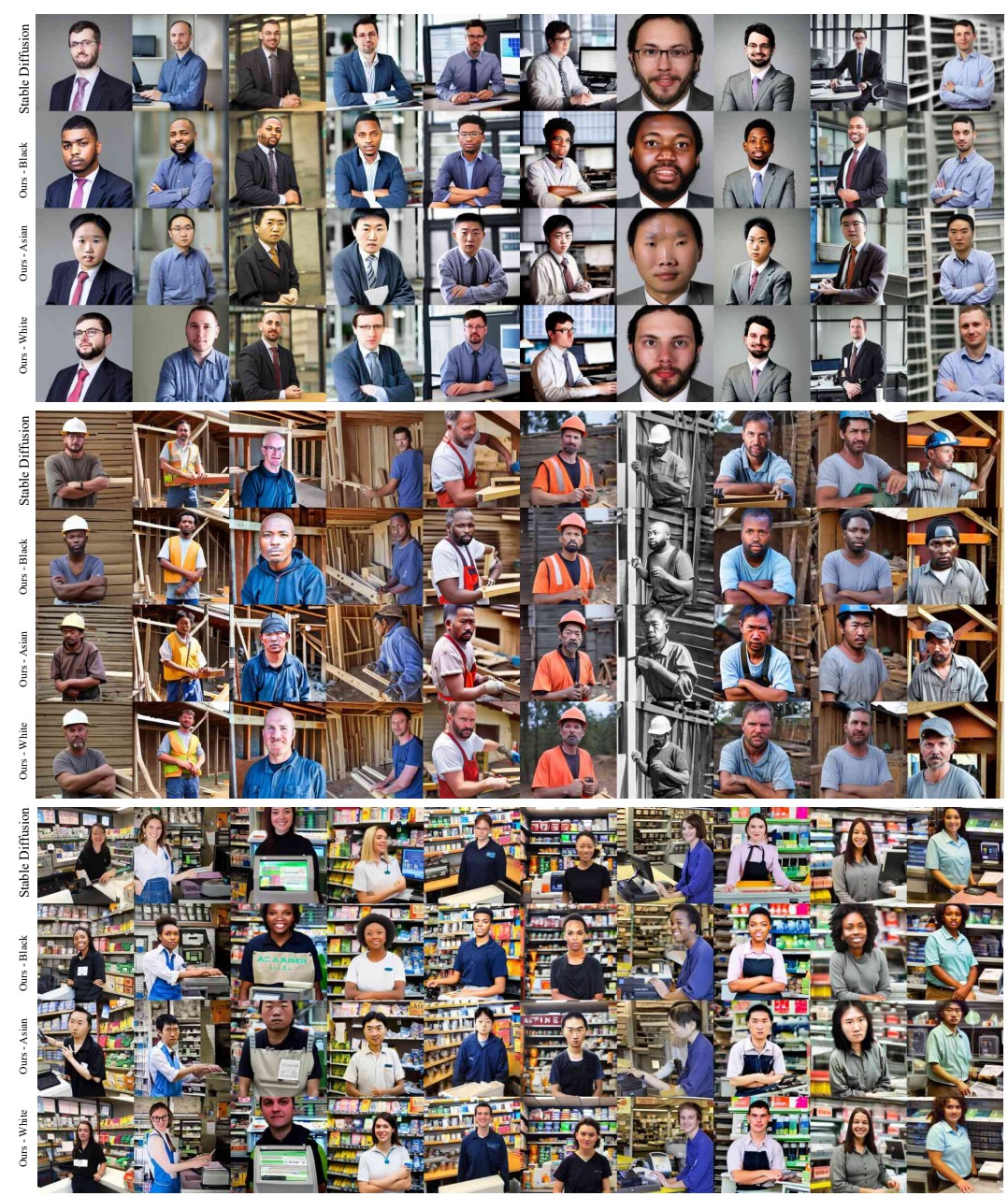

Figure 10: Comparison of RespoDiff and SD in generating images with respect to race (Black, Asian, White) across various professions, CEO, Construction worker, Cashier. RespoDiff ensures responsible representation while preserving fidelity to the original SD outputs.

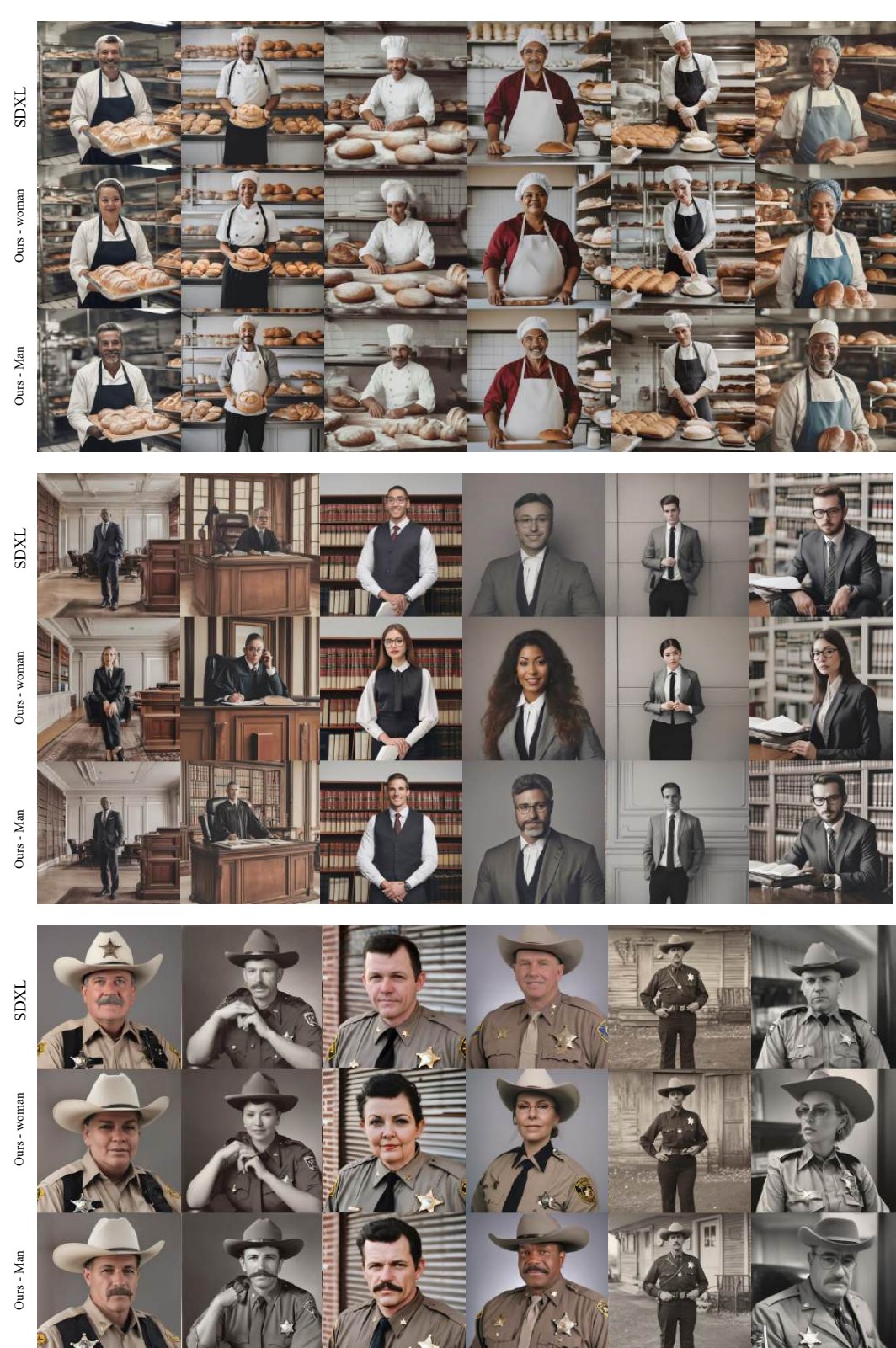

Figure 11: Comparison of RespoDiff and SDXL in generating images with respect to gender (Woman, Man) across various professions, Baker, Lawyer, Cook and Sheriff. RespoDiff ensures responsible representation while preserving fidelity to the original SDXL outputs.

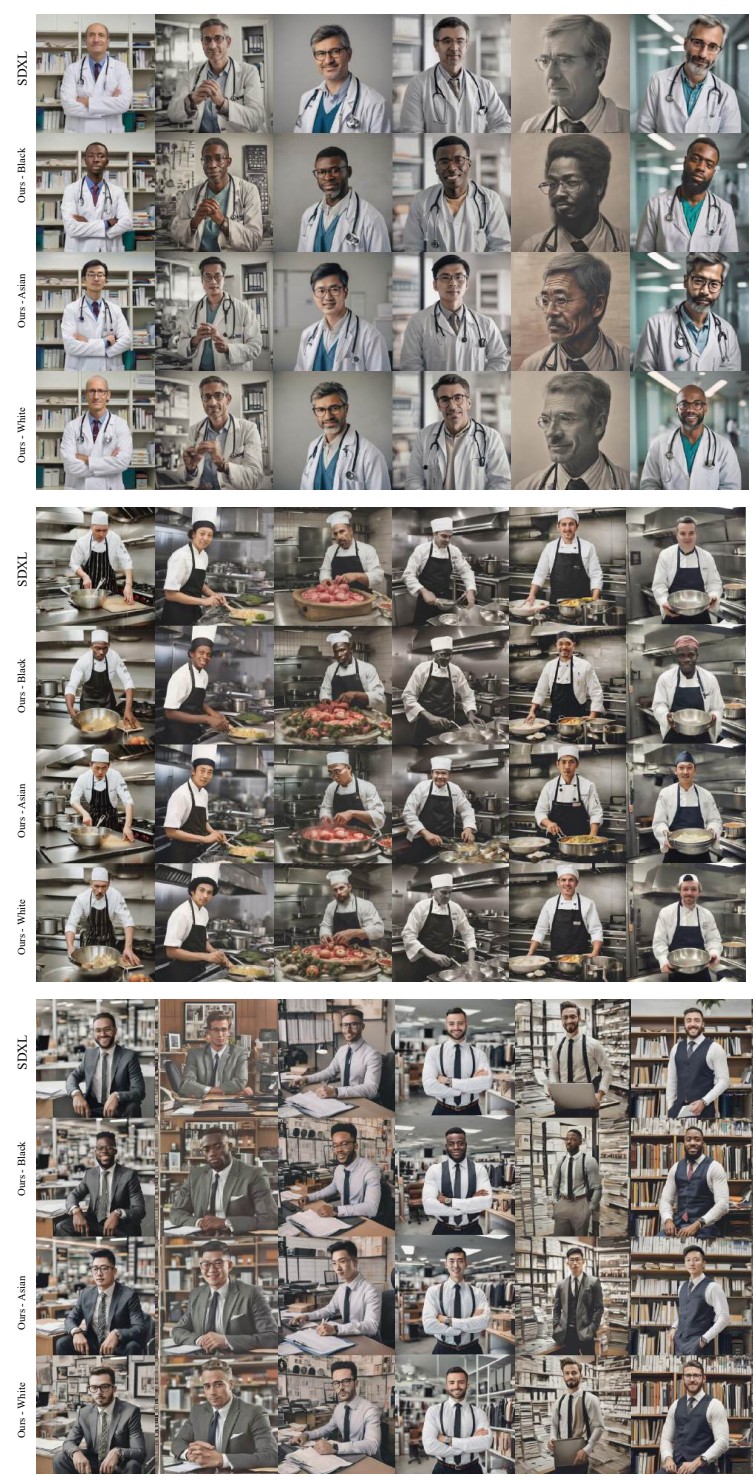

Figure 12: Comparison of RespoDiff and SDXL in generating images with respect to race (Black, Asian, White) across various professions, Doctor, Cook and Salesperson. RespoDiff ensures responsible representation while preserving fidelity to the original SDXL outputs.

