# OpenReview forum: "RespoDiff: Dual-Module Bottleneck Transformation for Responsible & Faithful T2I Generation"
_NeurIPS.cc/2025/Conference — NeurIPS 2025 poster_

### Official Review · Reviewer_skB6 · 2025-06-26

**Clarity:** 2
**Significance:** 3
**Originality:** 2
**Rating:** 4
**Confidence:** 4

**Summary:**

This paper introduces RespoDiff, a new framework designed to improve the fairness and safety of text-to-image diffusion models without compromising image quality or semantic accuracy. The core of RespoDiff is a dual-module transformation applied to the intermediate bottleneck representations of a diffusion model. To train these modules, the authors introduce a novel score-matching objective that effectively coordinates their actions.

**Questions:**

- **Q1**: The paper demonstrates its method using fixed, general neutral prompts like "a person" for fairness and "a scene" for safety. However, in a real-world deployment, the model would receive diverse and complex user queries (e.g., "a group of friends having a picnic"). For such queries, a fixed neutral prompt like "a person" might lose significant contextual information. How does the RespoDiff framework envision handling arbitrary user queries?
- **Q2**: For handling multiple concepts simultaneously (e.g., for intersectional fairness), the paper proposes a simple summation of the individual transformations. Have the authors investigated whether the effectiveness of one transformation change when combined with another?
- **Q3**: The paper introduces a crucial hyper-parameter, $\lambda$,  to weight the semantic loss. However, the paper does not discuss how this hyper-parameter was chosen. Could the authors provide an ablation study of the hyper-parameter?

**Ethical Concerns:**

["NO or VERY MINOR ethics concerns only"]

**Final Justification:**

My initial concerns centered on lack of clarity, novelty compared with the prior work SDisc, and practical applicability. However, the authors' rebuttal explanations and experiments effectively addressed these issues. Therefore, I decide to raise my score to 4.

**Limitations:**

yes

**Quality:**

3

**Strengths And Weaknesses:**

### Strengths

- **Balance between fairness/safety and fidelity**: Unlike previous methods that struggle with image quality, RespoDiff introduces the Semantic Alignment Module (SAM) and a score-matching objective to preserve visual quality and semantic coherence.

- **Thorough Evaluation**: The paper substantiates its claims with a thorough and rigorously designed evaluation.

### Weaknesses

- **Lack of Clarity**: While the paper describes RAM and SAM as transformation modules (e.g., main text in Section 4 and visual demonstrations in Figure 1), the actual implementation is an *input-agnostic* constant function that is linearly added. This is a bit overstatement and could cause confusion to normal readers.

- **Lack of Novelty**: Based on the previous point, the transformation module is essentially parameterized as a constant function, which is too naive. Furthermore, this architecture is almost identical to the previous work *SDisc* [Li et al., 2024].

- **Poor performance on different model**: While the method was applied to SDXL, the improvement in fairness was not as pronounced as with the v1.4 model, and it came at a higher cost to image fidelity. This suggests the method's effectiveness may not scale perfectly with newer, more complex model architectures.

- **Limited applicability**: See Questions Q1 and Q2.

---

> ### Author Rebuttal · Authors · 2025-07-30
>
> We thank the reviewer for the valuable feedback and thoughtful comments. We address each of the concerns below:
>
> ### **W1. Clarity**
> We present RAM and SAM as general transformation modules to keep the framework flexible and extensible. While our main experiments use constant vectors for simplicity and strong empirical performance, **our training framework and losses are architecture-agnostic**. As shown in Section 7.6 of the supplementary, we ablate alternative architectures such as MLPs and convolutional layers, and find that constant transformations work best in our setting. In short, RespoDiff’s contributions do not rely on any specific parameterization. RAM and SAM can be replaced with richer, input-dependent modules if desired, and our ablations confirm the method remains effective under such changes. We will clarify this in the final version.
> ### **W2. Novelty**
> We appreciate the reviewer’s concern. While it is true that our implementation uses constant transformation vectors similar to SDisc, our contributions are conceptually and technically distinct in several key aspects:
> - **Score-Matching Objective:** SDisc optimizes a reconstruction loss by generating target concept images, adding noise, and training a vector to denoise using the neutral prompt. In contrast, RespoDiff introduces a score-matching objective that explicitly models trajectory shifts between the neutral and target prompts using denoised latents. This provides more direct and structurally grounded supervision for learning responsible transformations. We highlight this difference in Line 104 of the main paper.
> - **Explicit semantic fidelity preservation:** A core novelty of our work is the Semantic Alignment Module, which explicitly regularizes the generated trajectory to remain close to the original diffusion model. This is an issue not addressed in SDisc, which lacks any such semantic fidelity constraint.
> - **Modular training:** We decouple concept steering (RAM) from semantic fidelity preservation (SAM) using dedicated score‑matching and semantic objectives with alternating optimization. This modular design is a key novelty of our approach and distinguishes it from SDisc’s single‑vector reconstruction setup.
> - **Empirical Performance:** RespoDiff consistently outperforms SDisc on fairness and safety metrics (Tables 1–2) while maintaining strong alignment. Qualitatively (Fig. 6; Sec. 7.4.5), SDisc often overfits to narrow stereotypes (e.g., sad or elderly faces, unrealistic profession depictions), likely due to its noise-reconstruction objective. In contrast, RespoDiff preserves profession context while effectively steering toward target concepts.
>
> We believe that the above elements constitute a distinct and novel framework relative to SDisc.
>
> ### **W3. Performance on SDXL**
> We would like to clarify that while the fairness gain with SDXL may appear less pronounced in absolute terms, it is still a substantial relative reduction (Deviation Ratio drops by **~64%** from 0.72 to 0.26). This is a significant improvement considering that SDXL is a more complex and larger model. Additionally, the slight increase in FID (+0.95) is marginal and still within high-quality generation bounds, especially when weighed against the large fairness gains. Importantly, the alignment scores (CLIP and WinoAlign) remain virtually unchanged or even slightly improved. To further validate RespoDiff on SDXL, we now include safety experiments showing that it effectively removes inappropriate content while preserving strong image fidelity.
>
> | Approach   | I2P (↓) | FID(30K) (↓) | CLIP(30K) (↑) |
> |------------|---------|------------|-------------|
> | SDXL       | 0.34    | 13.68      | 32.19       |
> | RespoDiff  | 0.17    | 13.90      | 32.10       |
>
> We also conduct additional experiments to investigate the effectiveness of our approach on a *non-UNet architecture such as Flux*. Due to resource and time constraints, we performed these experiments with Flux-mini. Since Flux uses transformer architectures without an explicit intermediate bottleneck space (as opposed to UNet architectures), we applied our modules directly to the text representations. Identifying a similarly interpretable latent space within transformers, as in UNet, required further exploration. We evaluated gender debiasing using this setup, and the results are summarized below.
>
> | Approach   | Dev Ratio (↓) | WinoAlign (↑) |
> |------------|----------------|---------------------|
> | Flux       | 0.71           | 24.73               |
> | RespoDiff  | 0.27           | 23.65               |
>
> The experimental results demonstrate that our approach can effectively enable responsible generation even on architectures like Flux. We will also provide the qualitative results for Flux in the final version of our paper. We believe this highlights the versatility of RespoDiff beyond UNet-based models. Overall, our work includes extensive evaluations across diverse model architectures, going beyond prior responsible generation methods, to support the robustness and generality of our approach.
>
> ### **Q1. Handling arbitrary user prompts**
> **Robustness to other neutral prompts:**  To assess the robustness of our approach to the choice of neutral prompt, and in response to the reviewer’s feedback, we conducted an additional experiment. We retrained the man and woman modules using an alternative neutral prompt “a group of people”, and evaluated them on two out-of-distribution prompts: “a group of friends on a picnic” and “a couple of doctors.” This allows us to test whether our method remains effective when trained on more complex neutral formulations. Due to the limitations of CLIP in capturing fairness and alignment for complex prompts, we employed GPT-4o for evaluation. We generated 200 images, uniformly sampling male and female modules, and used GPT-4o to:
> - Classify each image as male or female (to compute a deviation ratio).
> - Rate alignment to the prompt on a 1–5 scale. Images scoring ≥4 were considered well-aligned.
>
> | Prompt                         | Deviation Ratio (↓) | Alignment Accuracy (↑) |
> |-------------------------------|----------------------|-------------------------|
> | A group of friends in the picnic | 0.026               | 96.5%                  |
> | A couple of doctors             | 0.029               | 95.1%                  |
>
> Our results indicate that training with a different complex neutral prompt also yields balanced gender representations on arbitrary unseen queries such as ‘A group of friends in the picnic’,  while preserving prompt alignment, suggesting that the approach generalizes to other generic neutral prompts. We will present the qualitative results that support our analysis in the final version of our paper.
>
> **Handling arbitrary user queries:** As demonstrated in our experiments, our approach generalizes well across unseen contexts using a small set of general-purpose neutral prompts (e.g., “a person,” “a group of people”). To identify the neutral prompts, we envision training RespoDiff modules on a limited set of such prompts. At inference time, the system can select the most appropriate trained modules by computing similarity between the arbitrary user prompt and the available neutral prompts. For instance, a user prompt like “a group of friends on a picnic” would naturally align with a module trained on “a group of people.”
>
> ### **Q2. Intersectional fairness**
> In our intersectional fairness experiments (gender and race), we noticed some interference affecting gender more significantly than race. We believe this arises because we initially apply unit scales to both concepts. When composed, the concept with the larger effective residual can dominate, under‑steering the weaker one. To probe this, we varied per‑concept steering scales during composition (e.g., slightly >1 for gender and slightly <1 for race).
>
> **Evaluation of Gender fairness before and after Gender, Race composition**
>
> | Setting                          | Dev Ratio (↓) | WinoAlign (↑) |
> |----------------------------------|-----------|----------------|
> | Gender only                      | 0.14      | 27.30          |
> | Gender + Race (Scales 1, 1)      | 0.20      | 27.12          |
> | Gender + Race (Scales 1.1, 0.9)  | 0.15      | 27.09          |
>
> **Evaluation of Race fairness before and after Gender, Race composition**
>
> | Setting                          | Dev Ratio (↓) | WinoAlign (↑) |
> |----------------------------------|-----------|----------------|
> | Race only                        | 0.16      | 27.53          |
> | Gender + Race (Scales 1, 1)      | 0.14      | 27.12          |
> | Gender + Race (Scales 1.1, 0.9)  | 0.16      | 27.09          |
>
> As shown in the tables, light concept‑specific scaling rebalances the composition, restoring gender steering to near its single‑concept behavior without significantly changing alignment, while race fairness remains stable. We will include these results and qualitative examples in the final version. Additionally, learning the scaling factors dynamically to compose concepts is an interesting direction to explore in future work.
> ### **Q3. Sensitivity to $\lambda$**
> We now present results on the sensitivity of RespoDiff to the hyperparameter $\lambda$.
>
> | λ  | Dev Ratio (↓) |  WinoAlign (↑) | FID (30K) (↓) | CLIP (30K) (↑) |
> |------------|--------------|------------------|------------------|-------------------------|
> | 0          | 0.12         | 26.12            | 15.63            | 29.93                   |
> | 0.5 (Default) | 0.14      | 27.30            | 14.91            | 30.67                   |
> | 4          | 0.29         | 27.53            | 14.17            | 31.24                   |
>
> Empirically, increasing $\lambda$ keeps the trajectory closer to the base model, improving fidelity but weakening target‑concept steering, while decreasing $\lambda$ has the opposite effect. We therefore use $\lambda = 0.5 $ that balances fairness and fidelity.

---

> > ### Comment · Reviewer_skB6 · 2025-08-08
> >
> > Thank you for the rebuttal. The added explanations effectively clarify the differences between your work and SDisc, which helped highlight the paper's novelty. I am also convinced by the new experimental results, which demonstrate the practical utility of your method. Given these points, I have decided to increase my score.

---

> ### Author Response · Authors · 2025-08-08
> **Follow-up on Rebuttal - Request for Feedback from Reviewer skB6**
>
> Dear Reviewer skB6,
>
> We are reaching out to kindly initiate a discussion and request if you could take a moment to review our rebuttal and share any remaining concerns you might have. We have carefully addressed all comments raised by the reviewers, and the other three reviewers have responded very positively to our clarifications. If there is anything further we can clarify - beyond the concerns already discussed - we would be very glad to do so within the remaining author-reviewer discussion period. Thanks for your understanding.
>
> Best,
> The Authors

---

### Official Review · Reviewer_eyeL · 2025-06-30

**Clarity:** 1
**Significance:** 2
**Originality:** 2
**Rating:** 5
**Confidence:** 4

**Summary:**

The authors address the problem of responsible image generation, focusing on aspects such as gender, race, and safety. Their approach introduces two modules (RAM and SAM) which are activated after an initial t steps of generation using a neutral prompt. After this phase, training proceeds by alternating between the two modules: RAM is trained to steer the generation trajectory toward a more responsible prompt, while SAM is trained to preserve the content of the original prompt. The outputs of both modules are then combined.

The method is evaluated on Stable Diffusion 1.4 (SD1.4) and compared to existing techniques using metrics such as average deviation ratio (to assess fairness), prompt alignment, and image fidelity. An additional experiment was conducted on SDXL to test generalization beyond the baseline. An ablation study demonstrates the necessity of combining both RAM and SAM, rather than RAM alone.

**Questions:**

Q1) Would this work on a non-U-Net architecture, such as FLUX? This is the direction LDMs are moving.

Q2) Why is there no regularization to keep RAM small? If it is not a small offset from SAM, then why couldn't they just be the same?

**Ethical Concerns:**

["NO or VERY MINOR ethics concerns only"]

**Final Justification:**

Initially, my main concern was that the experimental models did not adequately reflect the current state of the field, which made it challenging to situate the work within contemporary literature and assess its contribution. However, the authors have now presented convincing results that better establish the relevance of their approach.

They have also clarified key aspects of their methodology and agreed to revisions that, in my view, significantly enhance the clarity of the manuscript.

With these primary concerns addressed, I now believe this paper presents a novel and effective approach to an important and timely topic.

**Limitations:**

yes

**Quality:**

2

**Strengths And Weaknesses:**

Strengths:

S1) The results in Tables 1 and 2 seem reasonable, on SD1.4.

S2) Table 4 shows promising preliminary results, suggesting that the method may also be effective on SDXL.

Weaknesses:

W1) The paper lacks a discussion of related work on diffusion models, which is essential for grounding the method within existing literature.

W2) Figure 1 is highly confusing. Labeling in the latent representations is inconsistent, and the transition from the initial t timesteps to later steps is unclear. The trapezoid shapes used for RAM and SAM suggest dimensionality reduction—if this is intended, it should be clarified. The arrows imply concatenation, yet the accompanying equations indicate element-wise addition. The red arrows for backpropagation are not intuitive, and the figure does not clearly depict the inference-time process.

W3) The term "SAM" is already widely used in the literature (e.g., Segment Anything Model), and its reuse here introduces unnecessary ambiguity.

W4) The choice of t, the step at which training shifts from the neutral prompt to RAM/SAM intervention, is neither justified nor explained.

W5) Sections 4.2 and 4.3 are mathematically dense and difficult to follow. The heavy use of in-line equations and nested symbols hinders readability and comprehension.

W6) The primary experiments are conducted on SD1.4, an outdated model with significantly lower visual quality compared to modern diffusion models. This raises concerns about the relevance and practical value of the results. The absence of core experiments on stronger models such as SDXL or FLUX is a major limitation.

W7)Tables 1 and 2 are poorly presented. The key takeaway (that fairness improves with minimal sacrifice to prompt alignment and image quality) is buried in the text. Without careful reading, the results may be misinterpreted as weak.

W8) The ablation study is incomplete. Specifically:

W8a) There is no analysis of the impact of varying t, the intervention start point.

W8b) It is unclear why RAM and SAM are trained in alternating fashion, rather than jointly. A comparison would strengthen the argument for this design choice.

W9) Between the method section and main figure, I found the method overall difficult to understand.

---

> ### Author Rebuttal · Authors · 2025-07-30
>
> We thank the reviewer for the constructive and thoughtful feedback. We will carefully revise the manuscript to address all writing-related concerns and improve overall clarity in the final version. Below, we respond to the remaining points in detail.
>
> ### **W1. Discussion of related work on diffusion models**
>
> Our primary focus is on enabling responsible (fair and safe) text-to-image generation rather than proposing changes to the diffusion process or diffusion models themselves.  Accordingly, we believe Sec. 3 of the main paper provides sufficient background on diffusion models and includes the most relevant related work to support a clear understanding of our approach. However, if the reviewer feels any important references have been overlooked, we greatly appreciate the suggestions and will be happy to include them in the final version.
>
> ### **W2. Regarding Figure 1**
>
> We clarify that the trapezoid shapes for RAM and SAM were chosen only to visually indicate transformations within the modules, not dimensionality reduction. Similarly, the arrows and red arrows were intended to simplify the visualization and avoid excessive complexity in the figure. We agree, however, that the current labeling can cause confusion and we will address these labeling inconsistencies clearly in the final version of the paper. Also, we will add a figure that depicts the inference-time process in the final version.
>
> ### **W4/W8a. The choice of $t$**
> We believe there may be a misunderstanding regarding RespoDiff's training process. In each training iteration, we sample a timestep $t$ randomly and update the shared RAM and SAM modules using a score-matching objective. These modules are not learned per timestep; instead, they are trained to capture average statistics across the entire diffusion trajectory. This random $t$ training choice is stated explicitly in the paper (lines 174 and 201). Accordingly, **the training intervention point is a randomly sampled $t$ at each iteration**. At inference, the same learned RespoDiff modules are applied at every sampling step to ensure fair and safe generation.  To further address the reviewer's concern regarding the choice of $t$, we conducted **targeted ablations by restricting the training-time sampling range**. Specifically, instead of sampling $t$ from all 1000 DDPM steps, we trained by sampling $t$ under two settings :
> 1. $t \in [600, 1000]$, representing noisier diffusion steps
> 2. $t \in [0, 350]$ , representing cleaner diffusion steps.
>
> The results of these controlled experiments are reported below.
>
> | Range of $t$         | Dev. ratio (↓) | WinoAlign (↑) |
> |---------------------|--------------:|------------------:|
> | 600–1000            | 0.08          | 26.12             |
> | 0–350               | 0.24          | 27.48             |
> | **0–1000 (default)**| 0.14          | 27.30             |
>
> It can be observed that the noisier steps $[600, 1000]$ reduce deviation to a large extent but severely hurt alignment; cleaner steps improve alignment but hurt fairness. Sampling across the full range offers the most balanced trade-off and is used in our main experiments. We will include this in the final version of our paper.
>
> ### **W6/Q1. Relevance of results on SD1.4 and adaptation to Flux**
>
> **Relevance of results on SD1.4 and additional safety results on SDXL :** We conducted the majority of our experiments and comparisons using Stable Diffusion v1.4 (SD v1.4), primarily because **prior responsible-generation methods (Li et al. 2024) have predominantly used SD v1-based models**. To ensure fair and meaningful comparisons, we adhered to the same setup. However, we recognize the importance of evaluating our approach on newer models. Therefore, **we provided core fairness results on SDXL as well**, reported in Table 4 and Table 5 of the main paper. We also provide qualitative analysis on SDXL in Figure 4, 10 and 11 to support our quantitative results. Since the existing baseline methods do not yet support SDXL, we compared our approach against the default SDXL baseline. Our results clearly demonstrate that RespoDiff effectively mitigates gender and racial biases in SDXL without compromising image fidelity. Furthermore, to substantiate our claims regarding the capability of RespoDiff to handle responsible generation using SDXL, we have now also conducted safety experiments on SDXL and present the corresponding results below. From the table, it can be observed that RespoDiff successfully eliminates inappropriate content while maintaining competitive image fidelity on SDXL.
>
> | Approach   | I2P (↓) | FID(30K) (↓) | COCO(30K) (↑) |
> |------------|---------|------------|-------------|
> | SDXL       | 0.34    | 13.68      | 32.19       |
> | RespoDiff  | 0.17    | 13.90      | 32.10       |
>
> **Adaptation to Flux:** Following the reviewer's suggestion, we conducted additional experiments to investigate the effectiveness of our approach on Flux. Due to resource and time constraints, we performed these experiments with Flux-mini. Since Flux uses transformer architectures without an explicit intermediate bottleneck space (as opposed to UNet architectures), we applied our modules directly to the text representations. Identifying a similarly interpretable latent space within transformers as in UNet, required further exploration. We evaluated gender debiasing using this setup, and the results are summarized below.
>
> | Approach   | Dev Ratio (↓) | WinoAlign (↑) |
> |------------|----------------|---------------------|
> | Flux       | 0.71           | 24.73               |
> | RespoDiff  | 0.27           | 23.65               |
>
> The experimental results demonstrate that our approach can effectively enable responsible generation even on architectures like Flux. We will also provide the qualitative results for Flux in the final version of our paper. We believe this highlights the versatility of RespoDiff beyond UNet-based models. Overall, our work includes extensive evaluations across diverse model architectures, going beyond prior responsible generation methods, to support the robustness and generality of our approach.
>
> ### **W8b. Joint training of RAM and SAM**
>
> During experimentation, we found that jointly training RAM and SAM using our combined loss led to higher deviation ratios and weaker alignment compared to our proposed alternating strategy. We believe this is due to gradient interference where the two modules may inadvertently oppose each other’s objectives, particularly in the early stages of training. In contrast, alternating training separates the updates: RAM is optimized solely with the concept alignment loss, and SAM is updated in a separate step using the semantic loss, conditioned on RAM’s output. This decoupling enables each module to specialize more effectively, reduces interference, and results in more stable optimization. We also explore a **related variant using a shared transformation in the supplementary material (Section 7.7, Table 15)**, where we again observe that alternating, modular updates lead to improved performance. We will include a direct comparison to joint training in the final version of the paper.
>
> ### **Q2. Regularization to keep RAM small**
>
> RAM serves as the steering component that shifts the latent representation toward the target responsible concept, while SAM acts as a regularizer that preserves alignment with the original model’s trajectory. We do not impose an explicit constraint to keep RAM small because (i) the required magnitude of transformation varies by concept and (ii) enforcing a small RAM norm could lead to under-steering, reducing the effectiveness of fairness or safety interventions.
> Instead, we regularize the composite transformation via the semantic loss (weighted by $\lambda$), which keeps the overall trajectory close to the original model.
>
>  **Shared transformation (Same module for RAM and SAM) :** We would like to clarify that the shared transformation design **has been evaluated in our supplementary material (Section 7.7; Table 15)**. Specifically, training a single transformation module using the combined loss underperforms compared to our proposed two-module setup (RAM + SAM) across all metrics. We believe this is because concept steering and semantic alignment are orthogonal objectives. When both are optimized through a single shared transformation, their competing gradients interfere, preventing effective specialization and resulting in suboptimal performance on both fairness and fidelity. In contrast, our modular design separates responsibilities: RAM applies strong, concept-specific transformations, while SAM maintains semantic consistency by remaining close to the identity. This decoupling enables each module to specialize effectively, resulting in improved performance across both dimensions. Additionally, such control is especially valuable for real-world deployment and interpretability, where the ability to tune each component independently is highly desirable. Achieving this level of modularity and transparency is challenging with a shared transformation approach.
>
> We thank the reviewer once again for their valuable feedback and hope that our responses have addressed all the concerns raised.

---

> > ### Comment · Reviewer_eyeL · 2025-08-01
> > **Rebuttal Response**
> >
> > I would like to thank the authors for their extensive efforts to address my concerns. With the clarifications and additional results, I see the paper as much stronger than the original version, and am inclined to raise my score. Specifically, the paper is 1) more clear and 2) better placed within current literature to me.
> >
> > For W2, W8b, and Q2, thank you for the clarification.
> >
> > **W1:** Given that the underlying problem revolves around these T2I generation models, I just find the sentence "T2I generation has transformed generative AI, enabling highly realistic image creation from text (Ho et al., 2020a; Ramesh et al., 2022; Rombach et al., 2022)" a bit lacking in setting the research context. Even small additions such as explicitly naming the models used later for the readers (SDXL, SDv1.4), or adding a few more SOTA models (e.g. FLUX, SD3) could be very helpful for providing the last bit of context for the reader. One sentence maximum would suffice.
> >
> > **W4/W8a:** You are correct that I was misunderstanding the choice of t a bit, and with your clarification, it all makes sense to me now.
> >
> > **W6/Q1:** Thank you for the clarification of continuing to include results comparable to older literature, and the choice to include results for SDXL brings the field forward. I believe the results on FLUX especially strengthen the paper, by showing the method is also applicable to transformer-based models.

---

> > > ### Author Response · Authors · 2025-08-01
> > > **Response to the Reviewer**
> > >
> > > We are glad that our rebuttal addressed the reviewer’s concerns and that the reviewer is now inclined to raise their score.
> > >
> > > In response to **W1**, we will include the following sentence immediately after the original:
> > >
> > > **W1**: “Models such as Stable Diffusion v1.4 (SDv1.4), SDXL, FLUX, and SD3 exemplify recent advancements in text-to-image (T2I) generation.”
> > >
> > > We believe this addition effectively addresses the concern. If the reviewer has any further questions, we would be happy to clarify them during the discussion phase.

---

### Official Review · Reviewer_oss3 · 2025-07-02

**Clarity:** 3
**Significance:** 3
**Originality:** 3
**Rating:** 4
**Confidence:** 4

**Summary:**

This paper proposes a framework for responsible and faithful text-to-image generation via dual-module bottleneck transformation in diffusion models. The method targets two critical yet often conflicting objectives: enhancing fairness/safety (e.g., reducing gender or racial bias, eliminating unsafe content) while preserving semantic fidelity to the input prompt. RespoDiff operates in the bottleneck latent space of a pre-trained diffusion model, introducing two learnable modules: Responsible Concept Alignment Module (RAM): steers generation toward fairness-aligned or safety-oriented concepts. Semantic Alignment Module (SAM): regularizes the transformation to maintain alignment with the original prompt's semantics.

**Questions:**

1. The training process assumes access to a well-defined neutral prompt. In practice, this may be subjective or culturally sensitive. Could the authors provide a strategy for automatically identifying or validating neutral prompts?

2. Can RespoDiff handle prompts that contain multiple overlapping responsible concepts (e.g., “a young Asian female doctor”)? Is there interference when applying multiple transformations, and if so, how is that managed?

3. Have the authors evaluated the method under ambiguous or adversarial prompts? How robust is RespoDiff to prompts that implicitly invoke bias?

**Ethical Concerns:**

["NO or VERY MINOR ethics concerns only"]

**Final Justification:**

The rebuttal and additional experiments provide strong empirical support for the method’s robustness and generality in the tested settings. While some opportunities remain for extending cultural/linguistic coverage, the key methodological concerns have been resolved to my satisfaction. Given these improvements, I am inclined to raise my score.

**Limitations:**

yes

**Quality:**

3

**Strengths And Weaknesses:**

RespoDiff's strength lies in the modular design that decouples ethical alignment from semantic preservation, enabling interpretable and reusable concept transformations.
However, two weaknesses remain. 1.RespoDiff relies on predefined, binary demographic concepts (e.g., "man" vs. "woman", or "Black" vs. "White"), limiting representation of non-binary concepts. 2, robustness to adversarial prompts or domain shifts is not explored. 3. the method assumes the availability of a neutral prompt, which may not always be easy to define in practice.

---

> ### Author Rebuttal · Authors · 2025-07-30
>
> We thank the reviewer for the valuable feedback and thoughtful comments. We address each of the concerns below:
>
> ### **W1. Reliance on predefined, binary demographic concepts**
>
> Our categories are not strictly binary - for race, consistent with prior works (Li et al., 2024), **we adopt three categories** (Black, Asian, White) as noted in Line 150 of the main paper. For gender, we currently consider man/woman; we acknowledge that this does not capture non‑binary identities and **explicitly note this limitation in Section 7.1**. We chose these categories to align with prior works (Li et al. 2024) and ensure a fair comparison.
> Regarding predefined concepts, we view responsible generation as comprising two complementary stages: (i) identifying the relevant fairness/safety concepts, and (ii) mitigating bias or unsafe content given those concepts. **Our work addresses (ii) mitigation**. Consistent with prior mitigation approaches, we assume the fairness/safety concepts are known in advance, which we acknowledged in the limitations. To the best of our knowledge, this is a common assumption in prior fair/safe generation works  (e.g., Li et al., 2024; Gandikota et al., 2024; Shen et al., 2024; Chuang et al., 2023). As noted in the main paper (Line 152), our categories and concepts are chosen to align with these works. That said, we believe that extending to unseen or emergent concepts is practical within the same framework since RespoDiff is modular. Each concept corresponds to a lightweight transformation. Hence, for any new concept, we can train an additional module without changing the backbone. Moreover, our approach can mitigate intersectional biases without additional training by composing existing transformations, as demonstrated in Section 7.4.9, which covers unseen combinations of known concepts and thus partially addresses “unseen” identities. However, we also agree that unseen/emergent concept discovery is important in practice and is orthogonal to our contribution. One approach would be to utilise a world‑model or LLM‑based procedure to propose new or emergent identities [C], after which RespoDiff can learn the corresponding transformations straightforwardly. We will make these points explicit in the main paper.
>
> [C]  D’Incà et al., OpenBias: Open-set Bias Detection in Text-to-Image Generative Models, CVPR 2024.
>
> ### **W2/Q3. Robustness to ambiguous prompts or domain shifts**
>
> **Regarding domain shifts:** Our modules are trained using general-purpose neutral prompts (“a person” for fairness, “a scene” for safety), but are evaluated on a wide variety of unseen prompts without any additional fine-tuning. For fairness, this includes profession-related prompts, while for safety, we use the I2P benchmark consisting of real-world, safety-critical queries, introducing a clear prompt distribution shift. Additionally, we assess image fidelity and alignment on the COCO-30K dataset, which spans diverse, non-human-centric domains. Across these varied settings, RespoDiff consistently improves fairness and safety metrics while preserving image quality (see Tables 1–3), demonstrating strong robustness to domain shifts. Moreover, our fairness evaluation is intentionally human-centric, as attributes such as gender and race are specifically defined for human domains.
>
> **Regarding ambiguous prompts that implicitly invoke bias:** Yes, we explicitly evaluate RespoDiff on ambiguous prompts that implicitly invoke bias, **as detailed in Section 7.4.7 of the supplementary**. Specifically, we consider prompts such as "a photo of a successful doctor" or "an image of a successful teacher," since terms like "successful" are known to amplify implicit gender biases in generation (Gandikota et al., 2024). Section 7.4.1 provides a detailed discussion on our prompt construction process. As shown in the results in Section 7.4.7, RespoDiff effectively mitigates such biases without degrading semantic fidelity, even in these challenging prompt scenarios.
> Additionally, we emphasize that our default evaluation setting, which uses professions from the WinoBias set (e.g., "doctor," "CEO"), already tests the model's robustness to implicitly biased prompts. Such prompts commonly trigger stereotypical gendered images (e.g., predominantly male for "CEO" or "doctor"), and addressing these implicit biases is precisely the focus of our mitigation approach
>
> ### **W3/Q1. Assumption on the availability of a neutral prompt**
> For fairness in human subjects, we adopt the neutral prompt “a person,” which we find generalizes well across diverse human contexts. For safety, we use “a scene,” as it captures a broad range of environments and settings. As shown in Tables 1–3, both prompts effectively support generalization to unseen scenarios. To further assess the robustness of our approach to the choice of neutral prompt, and in response to the reviewer’s feedback, we conducted an additional experiment. We retrained the man and woman modules using an alternative neutral prompt “a group of people”, and evaluated them on two out-of-distribution prompts: “a group of friends on a picnic” and “a couple of doctors.” This allows us to test whether our method remains effective when trained on more complex neutral formulations. Due to the limitations of CLIP in capturing fairness and alignment for complex prompts, we employed GPT-4o for evaluation. We generated 200 images, uniformly sampling male and female modules, and utilised GPT-4o to:
>
> - Classify each image as male or female (to compute a deviation ratio).
>
> - Rate alignment to the prompt on a 1–5 scale. Images scoring ≥4 were considered well-aligned.
>
> | Prompt                         | Deviation Ratio (↓) | Alignment Accuracy (↑) |
> |-------------------------------|----------------------|-------------------------|
> | A group of friends in the picnic | 0.026               | 96.5%                  |
> | A couple of doctors             | 0.029               | 95.1%                  |
>
> Our results indicate that training with a different complex neutral prompt also yields balanced gender representations on arbitrary unseen queries such as ‘A group of friends in the picnic’,  while preserving prompt alignment, suggesting that the approach generalizes to other generic neutral prompts. We will present the qualitative results that support our analysis in the final version of our paper.
>
> **Strategy for automatically identifying or validating neutral prompts:** As demonstrated in our experiments, our approach generalizes well across unseen contexts using a small set of general-purpose neutral prompts (e.g., “a person,” “a group of people”). To identify the neutral prompts, we envision training RespoDiff modules on a limited set of such prompts. At inference time, the system can select the most appropriate trained modules by computing similarity between the arbitrary user prompt and the available neutral prompts. For instance, a user prompt like “a group of friends on a picnic” would naturally align with a module trained on “a group of people.”
>
>
> ### **Q2. Handling prompts that contain multiple overlapping responsible concepts**
>
> RespoDiff can support prompts involving multiple responsible concepts by composing the corresponding modules at inference, without requiring additional training, **as detailed in Section 7.4.9**. In our intersectional experiments (gender and race), we noticed some interference affecting gender more significantly than race. We believe this arises because we initially apply unit scales to both concepts. When composed, the concept with the larger effective residual can dominate, under‑steering the weaker one. To probe this, we varied per‑concept steering scales during composition (e.g., slightly >1 for gender and slightly <1 for race).
>
> **Evaluation of Gender fairness before and after Gender, Race composition**
>
> | Setting                          | Dev Ratio (↓) | WinoAlign (↑) |
> |----------------------------------|-----------|----------------|
> | Gender only                      | 0.14      | 27.30          |
> | Gender + Race (Scales 1, 1)      | 0.20      | 27.12          |
> | Gender + Race (Scales 1.1, 0.9)  | 0.15      | 27.09          |
>
> **Evaluation of Race fairness before and after Gender, Race composition**
>
> | Setting                          | Dev Ratio (↓) | WinoAlign (↑) |
> |----------------------------------|-----------|----------------|
> | Race only                        | 0.16      | 27.53          |
> | Gender + Race (Scales 1, 1)      | 0.14      | 27.12          |
> | Gender + Race (Scales 1.1, 0.9)  | 0.16      | 27.09          |
>
> As shown in the tables, light concept‑specific scaling rebalances the composition, restoring gender steering to near its single‑concept behavior without significantly changing alignment, while race fairness remains stable. We will include these results and qualitative examples in the final version. Additionally, learning the scaling factors dynamically to compose concepts is an interesting direction to explore in future work.
>
> We thank the reviewer once again for their valuable feedback and hope that our responses have addressed all the concerns raised.

---

> > ### Comment · Reviewer_oss3 · 2025-08-04
> > **Comprehensive Response**
> >
> > I appreciate the authors’ detailed clarifications and additional experiments.
> >
> > - For neutral prompt identification/validation: The authors acknowledge the reliance on predefined neutral prompts and provide experiments showing that alternative prompts generalize well. They also outline a conceptual strategy for automatically selecting appropriate neutral prompts at inference via similarity matching. While this is a useful direction, it remains a proposed approach rather than a fully implemented and validated component, so the robustness of automated neutral prompt discovery is still not empirically demonstrated.
> >
> > - For handling multiple overlapping responsible concepts: The intersectional bias experiments show that light per-concept scaling can restore performance. This is a clear and satisfactory response that demonstrates the framework’s adaptability to multi-attribute settings.

---

> > > ### Author Response · Authors · 2025-08-04
> > > **Response**
> > >
> > > We thank the reviewer for the response. We are glad to have addressed the concerns on intersectional biases. Regarding the comment on automated neutral prompt discovery, we have now implemented the proposed similarity-based strategy for selecting neutral prompts at inference time. Specifically, we used CLIP’s text encoder to compute cosine similarities between user prompts and a small set of general-purpose candidate neutral prompts (e.g., “a person,” “human,” “a group of people”). We utilised both profession-based prompts (e.g., “A photo of a doctor”) from Winobias and prompts like “A couple of doctors,” “A group of friends on a picnic” as inference time (user) prompts. The results for a subset of user prompts are summarized below.
> > >
> > > | Target Prompt                     | a person | human  | a group of people |
> > > |----------------------------------|----------|--------|--------------------|
> > > | A photo of a Analyst             | **0.8682**   | 0.8215 | 0.8068             |
> > > | A photo of a Assistant           | **0.8753**   | 0.8352 | 0.8423             |
> > > | A photo of a Attendant           | **0.8461**   | 0.8086 | 0.7759             |
> > > | A photo of a Baker               | **0.8281**   | 0.7988 | 0.7854             |
> > > | A photo of a CEO                 | **0.8731**   | 0.8532 | 0.8387             |
> > > | A photo of a Carpenter           | **0.8321**   | 0.7952 | 0.7795             |
> > > | A photo of a Cashier             | **0.7791**   | 0.7426 | 0.7090             |
> > > | A photo of a Cleaner             | **0.8439**   | 0.8226 | 0.7950             |
> > > | A group of friends in picnic     | 0.7739   | 0.7361 | **0.8725**             |
> > > | A couple of doctors              | 0.8128   | 0.8023 | **0.8375**             |
> > >
> > > We observed that all 36 profession prompts from the Winobias dataset consistently showed the highest cosine similarity to the neutral prompt “a person,” while group-based prompts aligned most closely with “a group of people.” This demonstrates that the automated prompt discovery method would naturally select the same neutral prompts used in training, with which RespoDiff has already been shown to perform effectively for the user prompts considered, both in our main experiments ("a person") and the additional evaluations using an alternate prompt ("a group of people") presented in the rebuttal. These findings provide empirical support for the effectiveness of the inference-time prompt-matching strategy. We hope that our response has addressed the concerns raised.

---

> > > > ### Comment · Reviewer_oss3 · 2025-08-05
> > > >
> > > > The additional rebuttal directly addresses my earlier concern on automated neutral prompt discovery by actually implementing the proposed similarity-based strategy. The CLIP-based cosine similarity results clearly show that profession-based prompts map most closely to “a person” and group-based prompts to “a group of people,” aligning with the training prompts already shown to be effective. This provides concrete empirical support that the automated selection method would preserve performance for the tested scenarios. While broader cultural and linguistic coverage would further strengthen the claim, the new evidence substantially increases my confidence in the approach. I am therefore considering raising my score.

---

> > > > > ### Author Response · Authors · 2025-08-05
> > > > > **Response to Reviewer oss3**
> > > > >
> > > > > We are glad the additional experiments addressed your concerns. We truly appreciate your consideration to raise the score.

---

### Official Review · Reviewer_xWgu · 2025-07-02

**Clarity:** 4
**Significance:** 3
**Originality:** 3
**Rating:** 5
**Confidence:** 4

**Summary:**

This paper proposes RespoDiff, a dual-module transformation framework for improving the fairness and safety of text-to-image diffusion models. One module (RAM) steers generations toward responsible concepts such as demographic diversity or safe content, while the other module (SAM) aims to preserve consistency with neutral prompts. The paper introduces a score-matching objective to coordinate both modules and demonstrate improved results on fairness and safety benchmarks using both Stable Diffusion and SDXL.

**Questions:**

1.	Could you include comparisons to lightweight debiasing or editing methods, such as LoRA or adapter-based fine-tuning? This would help contextualize the added cost and complexity of your proposed method.
2.	How did you choose the training schedule (e.g., 1500 iterations with batch size 1), and did you observe stability or overfitting issues? Would larger batch sizes or early stopping change the results?
3.	Can you provide more comprehensive visual evaluation, such as class-wise breakdowns, distributional shifts, or qualitative failure cases, to complement the current figures?
4.	Are there strategies to improve FID when applying responsible transformations, or is some degradation unavoidable?
5.	What is your view on extending this method to support unseen or emergent concepts not included in the original training categories?

**Ethical Concerns:**

["NO or VERY MINOR ethics concerns only"]

**Final Justification:**

The authors effectively addressed the concerns from me (sensitivity of $\lambda$, and LoRA comparison, etc) and other reviewers, so I decide to raise the score.

**Limitations:**

No. The paper would benefit from a more direct discussion of its limitations. Specifically, the reliance on predefined concept sets and separate training per category introduces scalability and coverage concerns. Additionally, the societal impact of applying targeted transformations (e.g., selecting what constitutes a “safe” or “fair” output) should be considered more carefully, as these decisions may encode implicit normative judgments.

**Quality:**

3

**Strengths And Weaknesses:**

Strengths
1. Addressing bias and unsafe outputs in generative models is a critical challenge. The paper focuses on a practical and socially relevant goal: producing images that better reflect demographic balance and remove harmful content, without severely degrading fidelity. This target is timely given ongoing concerns with fairness in large-scale image generation.
2.	Splitting the model’s behavior into two learnable modules, target concept steering + preserving alignment, is a clean and modular approach. This separation makes the optimization more interpretable and offers a more principled alternative to ad hoc fine-tuning or prompt hacking. The score-matching formulation used to couple these modules also aligns with how diffusion models are typically optimized.
3.	RespoDiff outperforms existing methods such as SDisc, FDF, and BAct on deviation ratio and safety metrics across multiple datasets. For example, on the I2P benchmark, the model reduces the rate of unsafe generations significantly compared to SD and SLD, while maintaining competitive CLIP scores and acceptable image fidelity.

Weaknesses:
1. The paper briefly evaluates the individual contributions of RAM and SAM (Table 6), but stops short of deeply analyzing alternative designs or loss settings. For example, it would be helpful to evaluate whether a shared transformation performs substantially worse, or how sensitive the model is to the λ parameter balancing the two objectives.
2.	Each concept category (e.g., gender, race, safety) requires training transformations using thousands of steps. For instance, the safety module uses 1500 iterations with batch size 1. This fine-grained per-category training increases implementation overhead and raises questions about scalability. It would strengthen the work to compare with plug-and-play alternatives, such as LoRA or prompt reweighting, to show that the added complexity provides significant gains.
3.	The paper presents only a few examples (e.g., Fig. 2 and 3) to support visual performance claims. While illustrative, these are not enough to assess how consistently the method performs across prompt diversity. Additional aggregated qualitative results (e.g., distribution analysis, randomly sampled generations, user preference studies, or error analysis) would better support the fidelity and alignment claims.
4.	Although RespoDiff maintains semantic consistency and improves fairness, it introduces a drop in image fidelity, especially in the safety task (Table 3). For example, FID increases from 14.09 to 17.89 on COCO-30K when responsible transformations are applied. This suggests that there is still a trade-off between fairness and image quality, and it would be helpful to better understand or mitigate this.
5.	The model requires target concepts to be known ahead of time and does not generalize to new or intersectional identities outside of those categories (e.g., non-binary, mixed-race, or transgender). This predefined concept reliance is acknowledged briefly in the appendix, but not sufficiently addressed in the main paper. Broader or adaptive concept discovery would be needed to make the approach more flexible in practice.

---

> ### Author Rebuttal · Authors · 2025-07-30
>
> We thank the reviewer for the valuable feedback and thoughtful comments. We address each of the concerns below:
>
>
> ### **W1/Q1. Shared Transformation and Sensitivity to $\lambda$**
>
> **Shared Transformation:**  We would like to clarify that the shared transformation design **has been evaluated in our supplementary material** (Section 7.7; Table 15). Specifically, training a single transformation module using the combined loss underperforms compared to our proposed two-module setup (RAM + SAM) across all metrics. We believe this is because concept steering and semantic alignment are orthogonal objectives. When both are optimized through a single shared transformation, their competing gradients interfere, preventing effective specialization and resulting in suboptimal performance on both fairness and fidelity.  In contrast, our modular design separates responsibilities: RAM applies strong, concept-specific transformations, while SAM maintains semantic consistency by remaining close to the identity. This decoupling enables each module to specialize effectively, resulting in improved performance across both dimensions. Additionally, such control is especially valuable for real-world deployment and interpretability, where the ability to tune each component independently is highly desirable. Achieving this level of modularity and transparency is challenging with a shared transformation approach.
>
> **Sensitivity to  $\lambda$:** We now present results on the sensitivity of RespoDiff to the hyperparameter $\lambda$.
>
> | λ  | Dev Ratio (↓) | WinoAlign (↑) | FID (30K) ↓ | CLIP(30K) ↑ |
> |------------|--------------|------------------|------------------|-------------------------|
> | 0          | 0.12         | 26.12            | 15.63            | 29.93                   |
> | 0.5 (Default) | 0.14      | 27.30            | 14.91            | 30.67                   |
> | 4          | 0.29         | 27.53            | 14.17            | 31.24                   |
>
>
> Empirically, increasing $\lambda$ keeps the trajectory closer to the base model, improving fidelity but weakening target‑concept steering, while decreasing $\lambda$ has the opposite effect. We therefore use $\lambda =0.5$ as a knee point that balances fairness and fidelity.
>
> ### **W2. Comparison to LoRA**
>
> We now provide comparisons to LoRA-based approaches on the Doctor profession to test a plug‑and‑play alternative for gender debiasing. We trained two concept‑specific LoRAs (“male doctor” and “female doctor”) using SD v1.4 with the default HuggingFace training arguments (≈15k iterations per LoRA). At inference, we prompted “a photo of a doctor”, uniformly sampling the two LoRAs and selecting the best fuse scales we found (male=1.0, female=1.2).
>
> | Approach   | DevRat ↓ | Prompt Alignment ↑ |
> |------------|----------|---------------------|
> | LoRA       | 0.41     | 28.09               |
> | RespoDiff  | 0.03     | 28.15               |
>
> It can be observed that the LoRA approach produced a much higher deviation ratio than RespoDiff, while alignment was comparable. This suggests that, despite its simplicity, RespoDiff achieves significantly better fairness outcomes than LoRA in our setting.
>
> **Scalability:** While RespoDiff requires training (~1,500 steps) per concept per neutral prompt, this cost is incurred only once and is not tied to specific user prompts. For instance, modules trained using general prompts like “a person” (for fairness) or “a scene” (for safety) generalize effectively to diverse, unseen prompts without further finetuning, as demonstrated in our profession and I2P evaluations. Thus, we believe that the per‑concept cost is amortized over a wide prompt distribution.
>
> **Inference Overhead:** Compared to LoRA, RespoDiff introduces negligible overhead at inference. Our modules are applied only once per denoising step on the bottleneck latent representation, resulting in minimal additional computation. In contrast, LoRA typically modifies multiple cross-attention layers throughout the U-Net, which introduces repeated low-rank matrix operations and increases both memory and latency. In this regard, RespoDiff offers a plug-and-play mechanism for responsible generation with significantly lower runtime burden.
>
> ### **Q2. Choice of training schedule**
>
>
> We chose the schedule based on loss curves and qualitative analysis. Both the concept and semantic losses converge, and visualisations stabilize by ~1,500 iterations; training much longer (e.g., ≥10,000 iterations) introduced visible artifacts and overfitting to certain visual types. Larger batch sizes slightly increased the concept loss, but the qualitative results were essentially unchanged. Accordingly, we adopt 1,500 iterations with a small batch, which yields stable visuals with converged losses.
>
> ### **W3/Q3. Qualitative results**
>
>  Beyond Figs. 2–3 in the main paper, we provide extensive qualitative results in the supplementary: Sec. 7.8 (Figs. 8–9) presents SD v1.4 generations across a wide range of professions, Figs. 10–11 present corresponding results for SDXL, and Fig. 7 shows qualitative examples for safe generation. Across these diverse prompts, we believe RespoDiff reliably steers toward the intended target concept while maintaining visual fidelity. While we are unable to include additional qualitative analyses in the rebuttal phase due to the restriction on image submission, we appreciate the reviewer’s feedback and plan to incorporate further qualitative results, such as failure cases in the final version of the paper.
>
> ### **W4/Q4. FID for safety**
>
> We agree there is a trade‑off when enforcing safety. We believe this arises from the nature of safety‑oriented transformations and how FID is computed. Safety controls suppress potentially unsafe visual cues (e.g., weapon‑like shapes, exposed‑skin/texture patterns), even when such cues are faint or unintentional. Because these cues are entangled with broader scene/style attributes (lighting, background textures), the transformations can introduce distortions but systematic changes (extra coverage, local occlusions, background adjustments). These shifts move the generated distribution away from COCO’s statistics, which FID, measured in Inception feature space penalizes [A], even when semantic alignment remains high (consistent with our CLIP results in Table 3). By contrast, we believe that fairness‑oriented edits are typically localized identity adjustments (skin tone, facial cues) that preserve scene layout, lighting, and background statistics, so the induced distribution shift and thus the FID change is much smaller.  This behavior aligns with findings by Jayasumana et al. [B], where they show that FID increases with high distortions while being less sensitive to low distortions. Even with semantic regularization, some safety cases unavoidably move farther from the base distribution to remove unsafe cues, while keeping semantics intact. Within these constraints, our results indicate that RespoDiff improves safety more than prior methods while avoiding large fidelity degradation, yielding a comparatively stronger safety fidelity balance.
>
> [A] Jung et al. Internalized Biases in Fréchet Inception Distance NeurIPS 2021, DistShift Workshop
>
> [B] Jayasumana et al., Rethinking FID: Towards a Better Evaluation Metric for Image Generation, CVPR, 2024.
>
> ### **W5/Q5. Extension of RespoDiff to unseen concept categories/concept discovery**
>
> We appreciate this point. We view responsible generation as comprising two complementary stages: (i) identifying the relevant fairness/safety concepts, and (ii) mitigating bias or unsafe content given those concepts. Our work addresses **(ii) mitigation**. Consistent with prior mitigation approaches, we assume the fairness/safety concepts are known in advance, which we acknowledged in the limitations. To the best of our knowledge, this is a common assumption in prior fair/safe generation works  (e.g., Li et al., 2024; Gandikota et al., 2024; Shen et al., 2024; Chuang et al., 2023). As noted in the main paper (Line 152), our categories and concepts are chosen to align with these works. That said, we believe that extending to unseen or emergent concepts is practical within the same framework since RespoDiff is modular. Each concept corresponds to a lightweight transformation. Hence, for any new concept, we can train an additional module without changing the backbone. Moreover, our approach can mitigate intersectional biases without additional training by composing existing transformations, as demonstrated in Section 7.4.9, which covers unseen combinations of known concepts and thus partially addresses “unseen” identities. However, we also agree that unseen/emergent concept discovery is important in practice and is orthogonal to our contribution. One approach would be to utilise a world‑model or LLM‑based procedure to propose new or emergent identities [C], after which RespoDiff can learn the corresponding transformations straightforwardly. We will make these points explicit in the main paper.
>
> [C]  D’Incà et al.,OpenBias: Open-set Bias Detection in Text-to-Image Generative Models, CVPR 2024.
>
> We thank the reviewer once again for their valuable feedback and hope that our responses have addressed all the concerns raised.

---

> > ### Comment · Reviewer_xWgu · 2025-08-01
> > **Comprehensive and Solid Rebuttal**
> >
> > Thank you for your comprehensive and insightful rebuttal. I truly appreciate the effort you've put into addressing my concerns and providing additional experimental results. Your detailed responses have significantly alleviated most of my initial reservations and greatly strengthened my understanding and appreciation of your work.
> >
> > The added experiments (e.g., LoRA comparison, λ sensitivity) and clear explanations (e.g., FID trade-off, modular design, scope clarification) were particularly convincing. My confidence in your work has increased considerably.
> >
> > Most technical concerns are now fully resolved, and I strongly recommend integrating the key findings and explanations from this rebuttal directly into the main paper as you planned.
> >
> > In light of your excellent rebuttal, **I am pleased to raise my rating**.

---

> > > ### Author Response · Authors · 2025-08-03
> > > **Response to the reviewer**
> > >
> > > We are glad the rebuttal addressed your concerns. We truly appreciate your decision to raise the score and your encouraging feedback. We will ensure that the key clarifications are incorporated into the final version of the paper. If you have any remaining questions or suggestions, we would be happy to address them.

---

### Note · Authors · 2025-08-12

We would like to thank all the reviewers and the AC for their time, thoughtful evaluations, and constructive feedback throughout the review process. We also appreciate the reviewers’ recognition of our work in the following aspects:

- Relevance and importance of the problem (xWgu)
- Thoroughness of experiments and evaluations (xWgu, eyeL, skB6)
- Modular design and training objectives (xWgu, oss3, skB6)

Following the authors' rebuttal and extensive author-reviewer discussion, we believe that all reviewer concerns are comprehensively addressed and will be clearly reflected in the camera-ready version. We are also encouraged by the reviewers’ willingness to increase their scores. Below, we summarize the key revisions and clarifications to be made in response to reviewer feedback:

- Adding an ablation study on sensitivity to the $\lambda$ hyperparameter (xWgu, skB6)
- Including a comparison of RespoDiff with LoRA (xWgu)
- Providing additional qualitative results (xWgu)
- Clarifying the scope of RespoDiff and its extension to unknown concepts (xWgu, oss3)
- Evaluating robustness to alternate neutral prompts and discussion on automatic selection (oss3, skB6)
- Ablation on scaling effects during composition (oss3, skB6)
- Correcting labeling errors in figure (eyeL)
- Extending RespoDiff to transformer-based architectures (eyeL, skB6)
- Improving clarity in the discussion of T2I models in the introduction (eyeL)
- Addressing other writing-related concerns (eyeL)
- Adding a discussion on architecture-agnostic properties of RespoDiff (skB6)

We believe these revisions strengthen the clarity, scope, and contributions of our work. We look forward to incorporating these improvements in the final version.

---

### Decision · Program_Chairs · 2025-09-17

**Decision:**

Accept (poster)

**Comment:**

This paper proposes a framework for responsible and faithful text-to-image generation via dual-module bottleneck transformation in diffusion models. The proposed method addresses a critical challenge and outperforms existing baselines. All the reviewers lean to accept this paper. Therefore, I recommend accept. Please further revise the paper based on the reviewers' comments.